# Scenario Choice Impacts Carbon Allocation Projection at Global Warming Levels

Lee de Mora[1], Ranjini Swaminathan[2], Richard P. Allan[2], Jerry C. Blackford[1], Douglas I. Kelley[3], Phil Harris[3], Chris D. Jones[4], Colin G. Jones[5], Spencer Liddicoat[4], Robert J. Parker[6,7], Tristan Quaife[2], Jeremy Walton[4], and Andrew Yool[8]

[1]Plymouth Marine Laboratory, Plymouth, PL1 3DH, UK
[2]National Centre for Earth Observation and Department of Meteorology, University of Reading, Reading, UK
[3]UK Centre for Ecology & Hydrology, Wallingford, Oxfordshire, OX10 8BB, UK
[4]Met Office Hadley Centre for Climate Science and Services, Exeter, EX1 3PB, UK
[5]National Centre for Atmospheric Science, UK, and School of Earth and Environment, University of Leeds, Leeds, UK
[6]National Centre for Earth Observation, Space Park Leicester, University of Leicester, Leicester, UK
[7]Earth Observation Science, School of Physics and Astronomy, University of Leicester, UK
[8]National Oceanography Centre, European Way, Southampton, SO14 3ZH, UK

**Correspondence:** Lee de Mora (ledm@pml.ac.uk)

**Abstract.**

We show that the distribution of anthropogenic carbon between the atmosphere, land surface and ocean differs with the choice of projection scenario even for identical changes in mean global surface temperature. Warming thresholds occur later in lower carbon dioxide ($CO_2$) emissions scenarios and with less carbon in the three main reservoirs than in higher $CO_2$ emissions

scenarios. At 2 °C of warming, the mean carbon allocation differs by up to 63 PgC between scenarios, which is equivalent to approximately six years of the current global total emissions. At the same warming level, higher $CO_2$ concentration scenarios have a lower combined ocean and land carbon allocation fraction of the total carbon compared to lower $CO_2$ concentration scenarios.

The warming response to $CO_2$, quantified as the equilibrium climate sensitivity, ECS, directly impacts the global warming

level exceedance year and hence the carbon allocation. Low ECS models have more carbon than high ECS models at a given warming level because the warming threshold occurs later, allowing more emissions to accumulate.

These results are important for carbon budgets and mitigation strategies as they impact how much carbon the ocean and land surface could absorb at a given warming level. Carbon budgeting will be key to reducing the impacts of anthropogenic climate change, and these findings could have critical consequences for policies aimed at reaching net zero.

**Keywords:** Climate change, Carbon Cycle, Carbon Allocation, CMIP6, Earth System Models

## 1  Introduction

The Intergovernmental Panel on Climate Change's (IPCC) Sixth Assessment Report found that the global mean surface air temperature was 1.1°C warmer in the last decade (2011-2020) than in the pre-industrial era. They found that human activities

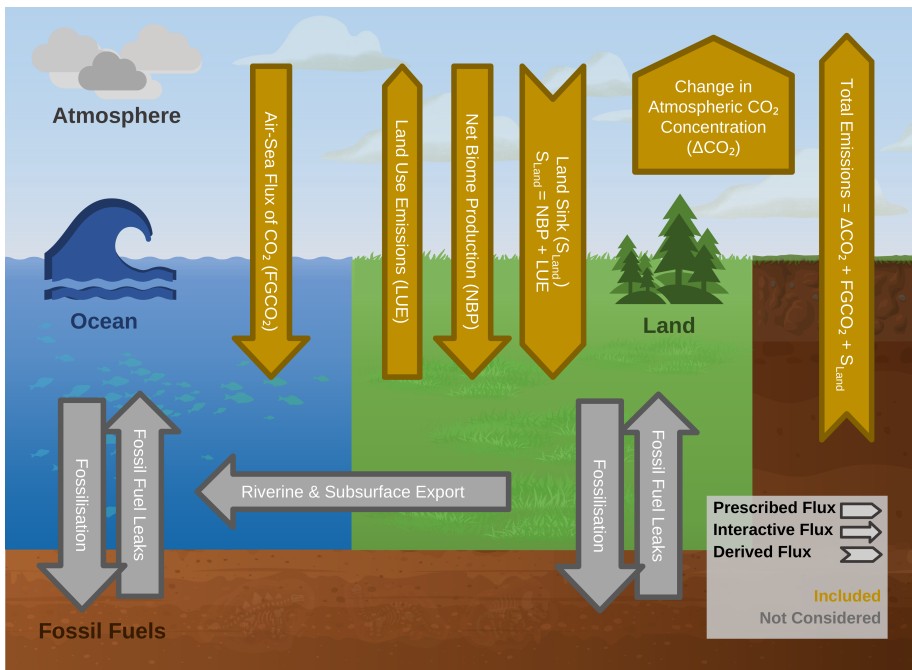

**Figure 1.** A simplified version of the Earth System carbon cycle. Interactive fluxes are shown as arrows, prescribed fluxes are shown as box arrows, and derived fluxes are shown as chevrons. The arrows in gold are considered in this analysis, and the grey arrows are not considered. The prescribed change in atmospheric carbon, $\Delta CO_2$, accounts for the anthropogenic fossil fuel exploitation and the subsequent carbon emission.

have indisputably caused this warming (IPCC, 2021b), with anthropogenic greenhouse gases, particularly carbon dioxide
($CO_2$), being the primary cause.

Since the industrial revolution, carbon has been transferred gradually from fossil fuel reservoirs to the atmosphere, primarily via combustion. Once in the atmosphere, some of the $CO_2$ is absorbed by the ocean via gas transfer, some is absorbed by the land surface via terrestrial carbon fixation, and some $CO_2$ remains in the atmosphere, as illustrated in Fig. 1. While these fluxes also occur naturally, the additional anthropogenic carbon load has perturbed the Earth System from its pre-industrial equilib-
rium. In the atmosphere, anthropogenic carbon causes additional warming (Hansen et al., 1981). In the ocean, anthropogenic carbon can cause acidification (Caldeira and Wickett, 2003) or participate in primary production or sequestration (Schlunegger et al., 2019). On the land surface, carbon can allow enhanced primary production and subsequent carbon sequestration. Once converted into biomass, this carbon may be a fuel source for fires which returns a portion of the sequestered carbon back to the atmosphere (Burton et al., 2022; Sullivan et al., 2022). Elevated atmospheric $CO_2$ can worsen food quality and nutrient
concentrations (Erda et al., 2005), and affect water-balance evapotranspiration, reducing streamflow in water-stressed regions (Ukkola et al., 2016).

The instantaneous distribution of anthropogenic carbon between the atmosphere, ocean and land surface is known as "carbon allocation in the Earth System" which we henceforth call "carbon allocation". The balance between these carbon sinks is hugely important to climate projections and policymakers (IPCC, 2021b), impacting warming feedbacks, marine biogeochemistry and life on land (Macreadie et al., 2019; Hilmi et al., 2021). The physical and biogeochemical feedbacks could affect the future rates of greenhouse gas accumulation in the atmosphere, directly impacting warming (Canadell et al., 2021). They also directly influence the remaining carbon budget, which policymakers may use to limit fossil fuel consumption in order to keep warming in line with policy goals (Jiang et al., 2021). In addition, the balance of carbon between the atmosphere, land and ocean has large-scale consequences on the future of climate engineering via $CO_2$ removal and solar radiation modification (Lawrence et al., 2018). Changes to carbon allocation also impact several United Nations Development Programme Sustainable Development Goals, notably 13: Climate Action, 14: Life below Water and 15: Life on Land (United Nations, 2015).

In observations, the atmospheric $CO_2$ concentration is typically measured directly, while the ocean and terrestrial $CO_2$ sinks are estimated with global process models constrained by observations. For the decade 2008–2017, the Le Quéré et al. (2018) synopsis of the global carbon budget summarised that the fossil fuel emissions were $9.4 \pm 0.5$ PgC yr$^{-1}$, and emissions from land use and land-use change was $1.5 \pm 0.7$ PgC yr$^{-1}$, most of which was due to deforestation. The growth of the atmospheric carbon was $4.7 \pm 0.02$ PgC yr$^{-1}$, the ocean carbon sink was $2.4 \pm 0.5$ PgC yr$^{-1}$, and the terrestrial carbon sink was $3.2 \pm 0.8$ PgC yr$^{-1}$. The difference between the estimated total emissions and the estimated changes in the atmosphere, ocean, and terrestrial biosphere was $0.5$ PgC yr$^{-1}$, which indicated that there were either overestimated emissions or underestimated sinks or both. There is also a flux of land carbon into the ocean via rivers between $0.45 \pm 0.18$ PgC yr$^{-1}$ and $0.78 \pm 0.41$ PgC yr$^{-1}$, but this flux is not generally included in recent models from the Sixth Coupled Model Intercomparison Project (CMIP6, described below in Sect. 1.1) (Jacobson et al., 2007; Resplandy et al., 2018; Hauck et al., 2020). There may also be a direct flux of fossil fuel extraction and other leaks into the ocean or land surface (Roser and Ritchie, 2023), but these are also neglected in models.

It is long established that the relationship between cumulative emissions and peak warming is insensitive to the emission pathway, either in the timing of emissions or the peak emission rate (Allen et al., 2009). More recently, Fig. 5.31 of Canadell et al. (2021) also supports negligible pathway dependence between the cumulative carbon emissions and the global mean temperature change in several projections with CMIP6 data.

The rising atmospheric $CO_2$ and warming climate will cause major changes in vegetation structure and function over large fractions of the global land surface. An increase in global land vegetation carbon has been projected, but with substantial variation between vegetation models (Friend et al., 2014). Much of the variability between models in global land vegetation carbon stocks was explained by differences in land vegetation carbon residence time (Jiang et al., 2015). In the ocean, the increase in atmospheric $CO_2$ enhances the ocean carbon storage while warming acts to decrease the ocean carbon storage (Katavouta and Williams, 2021).

Both the ocean and land carbon sinks are projected to continue to grow as the atmospheric concentration of $CO_2$ rises (Canadell et al., 2021). However, the combined fraction of emissions taken up by the land and ocean is projected to decline, and a larger fraction of the emissions will remain in the atmosphere. The carbon allocation at the year 2100 is strongly scenario

dependent (IPCC, 2021b, Fig. SPM.7). The projected atmospheric carbon allocation in the year 2100 ranges from 30% in SSP1-1.9 to 62% in SSP5-8.5. The Shared Socioeconomic Pathways (SSPs) are described below in Sect. 1.1. While the land and ocean carbon uptake are expected to remain approximately equal, the uncertainty is much larger in the land carbon sink than the ocean. The uncertainty in the land sink is due to the balance of carbon accumulation in the high latitudes against the loss of land carbon in the tropics, and the challenges of forecasting the water cycle, especially droughts, which significantly reduce the carbon absorption potential of the land surface (Ukkola et al., 2016; van der Molen et al., 2011; Canadell et al., 2021). On the other hand, continuous absorption of carbon into the ocean reduces its mean global buffering capacity and drives changes in the global ocean's carbonate chemistry, building a strong dependency on the choice of scenarios (Jiang et al., 2019; Katavouta and Williams, 2021).

## 1.1 Sixth Coupled Model Inter-comparison Project

Earth System models (ESMs) are the only tools capable of projecting a future coupled carbon-climate system. The Sixth Coupled Model Inter-comparison Project (CMIP6) is the most recent global effort to standardise, share and study ESM simulations (Eyring et al., 2016). The CMIP6 standard simulation protocols, called the Diagnostic, Evaluation and Characterization of Klima (DECK), are required simulations for a model to participate in CMIP6. The DECK includes a pre-industrial control, a gradual 1% $CO_2$ growth experiment and a rapid 4x$CO_2$ experiment. While it is not formally part of the CMIP6 DECK, models also contribute at least one historical simulation. For quality assurance, only models with a global drift of less than 10 PgC per century in the air-sea $CO_2$ flux and less than 0.1 °C per century in the volume mean ocean temperature are accepted (Jones et al., 2011; Eyring et al., 2016; Yool et al., 2020).

In order to make projections of the future anthropogenic climate drivers, multiple scenarios were proposed in the ScenarioMIP project to cover a wide range of potential futures (O'Neill et al., 2016). ScenarioMIP expands upon the CMIP6 DECK simulations with multiple scenarios of the future anthropogenic climate drivers that cover a wide range of potential future climate and human behaviours (O'Neill et al., 2016). Scenario names in CMIP6 ScenarioMIP are composed of one of the five shared socioeconomic pathways (SSP1-SSP5) followed by an estimate of the radiative forcing at the year 2100 between 1.9 and 8.5 Wm$^{-2}$. The five SSPs are narratives that describe broad socioeconomic trends that are expected to shape the future of humanity, and are based on trends in population, urbanisation, and technological and economic growth (Riahi et al., 2017). In this work, we include: two sustainable development scenarios SSP1-1.9 and SSP1-2.6; the intermediate emissions scenario, SSP2-4.5, which has a medium level of radiative forcing by the end of the century; the regional rivalry scenario, SSP3-7.0, which has more regional conflict and concerns about domestic security, pushing global collaboration into the background; and the enhanced fossil fuel development scenario, SSP5-8.5, which has extremely high fossil fuel deployment and atmospheric $CO_2$ concentration (O'Neill et al., 2016; Riahi et al., 2017).

## 1.2 Climate Sensitivity

Given the same rise in atmospheric $CO_2$ concentration, each ESM will warm to a different temperature due to the structural and parametric differences between models. The Equilibrium Climate Sensitivity (ECS) is defined as the global mean near-

surface air temperature rise in °C in response to a doubling of the atmospheric $CO_2$ concentration once the model has reached equilibrium. Constraints on the value of ECS are based on four lines of evidence: feedback process understanding, climate change and variability seen within the instrumental record, paleoclimate evidence, and emergent constraints (Arias et al., 2021). The 5-95% confidence range of ECS is between 2 °C and 5 °C, the likely ECS range is 2.5 - 4 °C, and the most likely value is 3 °C (Arias et al., 2021, TS.6). In ESMs, the spread in the sensitivity to $CO_2$ between models is one of the causes of uncertainty for when projections reach certain warming levels. Similarly, the uncertainty in the "allowable emissions" that would keep global temperature rise within policy targets are also impacted (United Nations Treaty Collection, 2015). This uncertainty is exacerbated in CMIP6 as it has a broader range of ECS values than previous generations and several CMIP6 models are outside the likely ECS range (Hausfather et al., 2022). Uncertainties in cloud feedbacks have been identified as the main cause of the large ECS range in CMIP6 (Ceppi and Nowack, 2021).

## 1.3 Global Warming Levels

Climate change policy has a tendency to focus on the climate at specific target years, such as 2050 or 2100 (United Nations Treaty Collection, 2015; IPCC, 2021a). However, due to the diversity of ECS values in CMIP6, an ensemble will project a wide range of warming rates and surface temperatures at a given point in time. This wide range of behaviours has knock-on effects on climate feedbacks and may distort the realism and representativeness of the ensemble's multi-model mean (Hausfather et al., 2022; Swaminathan et al., 2022). On the other hand, this more comprehensive range of responses is valuable in exploring carbon-climate processes that are of direct relevance to policy.

Instead of specific target years, we used the Global Warming Level (GWL) method (Swaminathan et al., 2022) to focus on three specific warming levels. These are 2 °C, 3 °C or 4 °C of warming relative to the pre-industrial period. The GWL method allows us to generate policy relevant assessments while exploiting the full ensemble of CMIP6 models. Not only does the GWL method mirror the policy discourse surrounding the warming targets, it is also largely independent of the choice of future emissions scenario as the world is thought to look generally the same at 2 °C, irrespective of the pathway (Hausfather et al., 2022). In addition, the GWL method bypasses the need to select or weight CMIP6 models as each model provides distinct and relevant information, so the full CMIP6 ensemble can be used (Hausfather et al., 2022). The three GWLs were chosen because the 2 °C GWL is a key target set in the 2015 Paris Agreement and thought to be a threshold for potentially dangerous climate change (United Nations Treaty Collection, 2015); the 3 °C GWL is close to the warming level that current nationally determined emission policies will realise for the year 2100 assuming a median climate sensitivity (United Nations Environment Programme, 2019); and finally, the 4 °C GWL is a low likelihood but high impact outcome if climate sensitivity is higher than the median estimate or emission reductions and climate policy break down (World Bank, 2012).

This is the first work that presents the carbon allocation using the GWL method. Previous analyses project carbon allocation at an arbitrary point in time using the mean of a set of models with widely different warming rates and sensitivities (IPCC, 2021b; Canadell et al., 2021). When compared against projections at specific points in time, our results are less influenced by the overall climate sensitivity of the ensemble and may be more relevant for policymakers.

## 2 Methods

### 2.1 Carbon allocation calculation

We calculate the carbon allocation for the land, ocean and atmospheric reservoirs separately. The amount of carbon in the land sink, $S_{Land}$, is derived from two other fields: the net biome production, *NBP*, and global total land use emissions, *LUE*, as shown in Fig. 1. The *NBP* is a diagnostic variable calculated by the models and is defined as positive for fluxes into the land carbon store in CMIP6 (Jones et al., 2016). $S_{Land}$ is the activity of the vegetation, which is the combined carbon flux of all natural sources, including photosynthesis, respiration, wildfire and other sinks and sources. These natural fluxes and therefore

the carbon sinks are altered by anthropogenic carbon emissions into the atmosphere, for example from fossil fuel combustion. $S_{Land}$ is positive in the direction of a sink into the land from the atmosphere, but it does not include the effects of anthropogenic land-use change. The *LUE* are anthropogenic carbon emissions, including deforestation, land management, reforestation and others (Lawrence et al., 2016). *LUE* is positive into the atmosphere. *NBP* is a diagnostic that combines both $S_{land}$ and *LUE*. *NBP* is positive into the land, so for these sign conventions, $NBP = S_{Land} - LUE$, and represents the net exchange between

land and atmosphere including anthropogenic emissions relating to land use change. The directions of these fluxes that are taken as positive as indicated in Fig. 1. To diagnose only the $S_{Land}$ component, it is therefore necessary to add back in the *LUE* to *NBP*. As such, $S_{Land}$ here is computed as the sum of the global total net biome production and the global total land use emissions:

$$S_{Land} = NBP + LUE \tag{1}$$

ESMs produce *NBP* as a diagnostic field in the *nbp* dataset, but this is actually their total carbon change in the land. It is not possible to directly isolate the *LUE* for each model and ensemble member in CMIP6 simulations, and the *LUE* values are calculated from prescribed land use scenarios and are common across all models and all ensemble members following Liddicoat et al. (2021). A more accurate method of determining the *LUE* would be to calculate the difference in net biosphere production between a pair of simulations, one with land use changing over time, and the other with fixed land use (Pongratz

et al., 2014; Liddicoat et al., 2021). However, these simulation pairs exist only for a limited subset of models and scenarios as part of the Land Use Model Inter-comparison Project, LUMIP (Lawrence et al., 2016). In practice, we calculated the global total net biome production as the cumulative sum along the time axis of the land surface *NBP* multiplied by the cell surface area then summed with the annual *LUE* value from Liddicoat et al. (2021).

The ocean component of the carbon allocation, $S_{Ocean}$, is the total global sum of the air-sea flux of $CO_2$. We calculated this

as the sum of the air-sea flux of $CO_2$ multiplied by the ocean area of each cell, expressed as a cumulative sum of the annual totals. Like the land surface, the ocean can be both a sink and a source of $CO_2$.

In the atmosphere, the global mean $CO_2$ concentration is provided in the scenario forcing from ScenarioMIP in units of parts per million (ppm). The total mass of the carbon in atmospheric $CO_2$, $C_{Atmos}$, is calculated by multiplying the change in concentration relative to the 1850 value in ppm by a constant factor. This conversion factor is 2.13 PgC per ppm change in

$CO_2$ concentration (Myers, 1983). No matter how much carbon the land and ocean components absorb from the atmosphere,

the atmospheric concentration of $CO_2$ will always strictly follow the prescribed atmospheric $CO_2$ concentrations of the forcing scenario. This means that anthropogenic emissions differ for each model but can be estimated (Jones et al., 2013). The total anthropogenic carbon, $C_{Total}$, is the sum of the total $CO_2$ in the atmosphere, $C_{Atmos}$, the total global sum of the $CO_2$ flux into the sea, $S_{Ocean}$, and the land sink, $S_{Land}$:

$$C_{Total} = C_{Atmos} + S_{Ocean} + S_{Land} \qquad (2)$$

## 2.2 Models Included in this Study

This analysis used all CMIP6 ESMs for which the following three variables were available as monthly averages over the time period 1850-2100: the near-surface atmospheric temperature (*tas*), the net biome productivity (*nbp*) and the air to sea flux of $CO_2$ (*fgco2*). For each scenario, we limited the size of the ensemble for each model to ten members, and required at least one historical and future scenario pair for each ensemble member. The grid cell area was also required for the ocean (*areacello*), and for land and atmosphere (*areacella*) grids. We excluded the entire ensemble member if any variables were absent, the time series was incomplete, or the data could not be made compliant with CMIP6 standards.

Each modelling centre has flexibility on which scenarios they simulate and how many ensemble members are generated for each scenario. This means that there is wide variation in the number of ensemble members between models. For instance, the UKESM1-0-LL model produced 19 different variants for the historical experiment, each using slightly different initial conditions drawn from the pre-industrial control simulation (Sellar et al., 2020). This generates an ensemble of variants which samples a wide range of the unforced variability simulated by the model. By spanning the range of internal variability, the mean of a single model ensemble can give a more robust estimate of its forced climate change response. In our work, we used a "one model - one vote" weighting scheme to balance models with large ensembles against models with small ensembles. This ensured that each model was given equal weight in the final multi-model mean. In practice, each ensemble member of a given model was weighted inversely proportional to the number of ensemble members that the model contributed. For reasons described in Sect. 1.3, we did not weight the results on the basis of the model quality, sensitivity or historical performance.

Table 1 lists the contributing models, the number of ensemble members for each scenario, and each model's equilibrium climate sensitivity (ECS). The ECS plays a first order role in how rapidly a given model reaches a given GWL for a given $CO_2$ pathway. For most models, we took the ECS value from Zelinka et al. (2020). For the models whose ECS was not included in Zelinka et al. (2020), we use the following ECS values: ACCESS-ESM1-5 from Ziehn et al. (2020), CMCC-ESM2 from Lovato et al. (2022), EC-Earth3-CC from Hausfather et al. (2022), GFDL-ESM4 from Dunne et al. (2020), and MPI-ESM1-2-LR from Mauritsen et al. (2019). No ECS value was available for the CanESM5-CanOE model as results were not available for the abrupt 4x$CO_2$ experiment required to calculate ECS using the Gregory method (Gregory et al., 2004; Christian et al., 2022). However, it only differs from CanESM5 by the addition of a marine biogeochemistry component model (Swart et al., 2019; Christian et al., 2022). We follow the method used elsewhere (Hausfather et al., 2022; Scafetta, 2022), and use CanESM5's ECS value for CanESM5-CanOE. Other ECS datasets also exist, see for instance: Flynn and Mauritsen (2020); Meehl et al. (2020); Weijer et al. (2020); Hausfather et al. (2022), and agree to less 0.1 °C with the values used in this study. All ECS values included here were derived using the Gregory et al. (2004) method; we note, however that the value of ECS for any given

model is sensitive to the method that was used to derive it. See for instance Table 4 of Boucher et al. (2020), where ECS for the same model may vary by more than 1 °C depending on the methodology.

The last row of Table 1 shows the ensemble mean ECS of the contributing models for each scenario. Following the "one-model one-vote" scheme, the "weighted ECS" only takes into account the presence or absence of models, not the number of contributing ensemble members. The spread of weighted ECS values between scenarios is small, ranging from 3.96 for SSP1-1.9 to 4.17 for SSP5-8.5. Five out of six of these ensemble means sit above the likely ECS range of 2.5 - 4 °C, and four of the individual models are outside the 5-95% confidence band, 2 - 5°C (Sherwood et al., 2020; Arias et al., 2021).

As seen in other CMIP ensemble studies, we attempt to maximise the number of models in this work in order to improve robustness (Flynn and Mauritsen, 2020; Meehl et al., 2020; Weijer et al., 2020; Hausfather et al., 2022). This means that we include all available candidates, even pairs of sibling models: thus, there are two CESM2 models and two CanESM5 models in the ensemble. CESM2-WACCM6 is configured identically to CESM2, except that it has expanded aerosol chemistry and uses 70 vertical levels and its model top is at $4.5 \times 10^{-6}$ hPa (approximately 130 km), instead of CESM2's 32 vertical levels and a model top at 2.26 hPa (approximately 40 km) (Danabasoglu et al., 2020). As noted above, the CanESM5-CanOE model differs from CanESM5 by the addition of a more complex marine biogeochemistry component (Christian et al., 2022).

In addition to sibling models, the same individual component models are used by several modelling centres. For instance, the NEMO ocean circulation model forms the marine circulation component model of six of the ESMs used here (Heuzé, 2021). While the ESMs use differing versions of NEMO with different configurations and settings, these models could not strictly be treated as statistically independent. However, it is beyond the scope of this work to develop or apply a method to weight models such that the multi-model mean is statistically robust, as in for instance, Brunner et al. (2020).

## 2.3   Global warming level calculation

We calculated the global warming level following the method of Swaminathan et al. (2022). The global mean atmospheric surface temperature is calculated for each model, scenario and ensemble member. The anomaly is the difference from the mean of the 1850-1900 period from the relevant historical ensemble member. This temperature time series is then smoothed by taking the mean of a window with a width of 21 years, i.e. 10 years either side of the central year. The first year that the smoothed global mean surface temperature anomaly exceeds the global warming level is the GWL exceedance year, as in Fig. 1 of Swaminathan et al. (2022). Due to the size of our smoothing window and the fact that these simulations end in 2100, the last possible GWL exceedance year is 2090.

We calculate the multi-model mean for each of the variables using the "one model - one vote" scheme described above. We also determine the multi-model mean GWLs and their exceedance years from the multi-model mean temperature, instead of taking the weighted mean of the GWL exceedance years for each ensemble member. This method ensures that the multi-model mean is more representative of the overall ensemble, instead of being biased towards only those models that reach the GWL.

We used the ESMValTool toolkit to perform our analysis. ESMValTool is built to facilitate the evaluation and inter-comparison of CMIP datasets by providing a set of modular and flexible tools (Righi et al., 2020). These tools include quick ways to standardise, slice, re-grid, and apply statistical operators to datasets. In our case, we used the `annual_statistics`

**Table 1.** A list of the models, the number of contributing ensemble members for each scenario, the model ECS, and the weighted mean ECS of the contributing models.

| Model | Historical | SSP1-1.9 | SSP1-2.6 | SSP2-4.5 | SSP3-7.0 | SSP5-8.5 | ECS |
|---|---|---|---|---|---|---|---|
| ACCESS-ESM1-5 | 3 | | 2 | 3 | 2 | 1 | 3.87 |
| CESM2 | 3 | | 3 | 3 | 3 | 3 | 5.15 |
| CESM2-WACCM | 3 | | 1 | 3 | 1 | 3 | 4.68 |
| CMCC-ESM2 | 1 | | | 1 | | | 3.57 |
| CanESM5 | 10 | 10 | 10 | 10 | 10 | 10 | 5.64 |
| CanESM5-CanOE | 2 | | 2 | 2 | 2 | | 5.64 |
| EC-Earth3-CC | 8 | | | 8 | | 1 | 4.23 |
| GFDL-ESM4 | 1 | 1 | 1 | 1 | 1 | 1 | 2.7 |
| IPSL-CM6A-LR | 12 | 5 | 3 | 6 | 10 | 5 | 4.56 |
| MIROC-ES2L | 5 | 5 | 5 | 5 | 5 | 5 | 2.66 |
| MPI-ESM1-2-LR | 5 | 5 | 5 | 5 | 5 | 5 | 2.83 |
| NorESM2-LM | 2 | | 1 | 2 | 1 | | 2.56 |
| UKESM1-0-LL | 10 | 5 | 10 | 10 | 10 | 5 | 5.36 |
| Total number of Ensembles | 65 | 31 | 43 | 59 | 50 | 39 | |
| Total number of Models | 13 | 6 | 11 | 13 | 11 | 10 | |
| Weighted ECS | 4.11 | 3.96 | 4.15 | 4.11 | 4.15 | 4.17 | |

preprocessor to calculate the annual mean, the `mask_landsea` preprocessor to mask the land or ocean areas, and the `area_statistics` preprocessor to calculate the area weighted global mean. ESMValTool is hosted on GitHub, and we have made available all of the code used in the study (see Code and data availability section).

## 3  Results

### 3.1  Multi-model mean carbon allocation

The total multi-model mean carbon allocation for each scenario in the year 2100 and for each of the three GWLs is shown in Fig. 2. The top pane shows the carbon allocation for the year 2100. At 2100, the higher emission scenarios have greater total carbon allocations and more of that carbon is allocated to the atmosphere, relative to the lower emission scenarios. At the year 2100, more carbon is allocated to the ocean than the land in SSP5-8.5, SSP3-7.0 and SSP2-4.5, while more carbon is allocated to the land than the ocean in SSP1-1.9 and SSP1-2.6. This reproduces the results from Fig. SPM.7 of IPCC (2021b) as discussed earlier in Sect. 1.

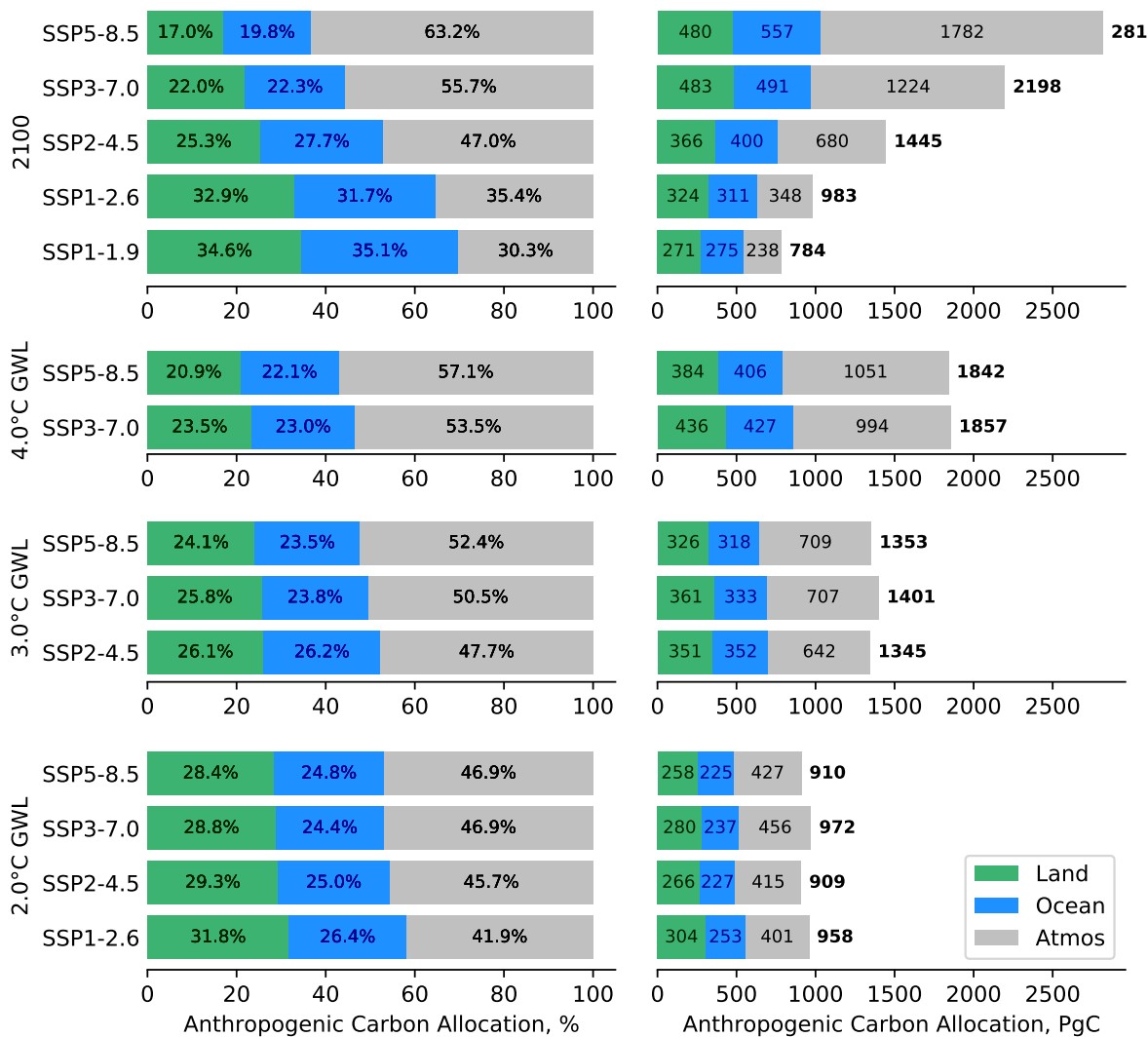

**Figure 2.** Carbon allocation for the multi-model mean for each scenario for the year 2100 and the three GWLs. The green, blue and grey areas represent the land, ocean and atmospheric carbon allocations. The left side shows the percentage allocation, and the right side shows the totals in PgC. The total values are shown in bold to the right of the bars. These values are rounded to the nearest 0.1% or the nearest integer PgC, so the three values may not add exactly to 100% or the total.

The lower three panes of Fig. 2 show the carbon allocation at each GWL. In all cases, the variability between scenarios at a given GWL is significantly less than the variability between scenarios at the year 2100 in the top pane. However, the variability within the same GWL is still significant in absolute terms. For instance, the multi-model mean total carbon allocation for the 2 °C GWL ranges from 909 PgC in SSP2-4.5 to 972 PgC in SSP3-7.0 (a range of 63 PgC). At the 3 °C GWL, the range is 56 PgC and at 4 °C GWL, the range is 15 PgC. When compared against the annual total emissions estimate, $9.4 \pm 0.5$ PgC yr$^{-1}$

(Le Quéré et al., 2018), these differences between scenarios represent several years' worth of the global total anthropogenic emissions.

In the land surface, the multi-model means have a range of 46 PgC, 35 PgC and 52 PgC between scenarios for the 2 °C, 3 °C, 4 °C GWLs respectively. The recent annual terrestrial carbon sink was $3.2 \pm 0.8$ PgC $yr^{-1}$ (Le Quéré et al., 2018), so the difference between scenarios is equivalent to at least a decade's worth of current carbon absorption by the land surface.

The multi-model means of the ocean flux have a range of 28 PgC, 34 PgC and 21 PgC between scenarios for the 2 °C, 3 °C, 4 °C GWLs respectively. This reflects the previous result that the carbon allocation to the land surface is more variable than the ocean, as the land values have wider ranges. The most recent annual estimate for the ocean carbon sink is $2.4 \pm 0.5$ PgC $yr^{-1}$ (Le Quéré et al., 2018). As in the case of the land described above, the difference between scenarios is equivalent to approximately one decade's worth of the current ocean carbon absorption.

In the left hand side of Fig.2, the higher $CO_2$ concentration scenarios have a larger atmospheric fraction than lower $CO_2$ concentration scenarios at the same GWL. For instance, the atmospheric fraction is 46.9% in SSP5-8.5 and 41.9% in SSP1-2.6 at the 2 °C GWL, and the atmospheric fraction is 52.4% in SSP5-8.5 and 47.4% in SSP2-4.5 at the 3 °C GWL.

Figure 2 only shows the multi-model means, not results for individual models; so the multi-model means that do not reach the GWL are not included in this figure. Table 1 shows that there are six models contributing to the SSP1-1.9 scenario in
this analysis, yet the multi-model mean does not reach the 2 °C GWL here. Similarly, there are 11 SSP1-2.6 models, but the multi-model mean does not reach the 3 °C GWLs before the year 2100, nor does the mean of the 13 SSP2-4.5 models reach 4 °C of warming.

## 3.2 Carbon allocation time series

The CMIP6 multi-model mean carbon allocation time series is shown in Fig. 3. The top left pair shows the development over
the historical period and the other five pairs show the projections. We include all data cumulatively from the year 1850, and all the cumulative carbon panes share the same y-axis range. The exceedance years for each of the multi-model mean GWLs are marked as vertical lines.

In the historical pane of Fig. 3, the fractional atmospheric carbon starts to grow in the second half of the 20th century, as the land fraction declines and the ocean fraction increases. However, all three reservoirs increase in absolute terms over the
entire historical period. By the end of the historical period, the land and ocean match reasonably well against the observational records of Raupach et al. (2014) and Watson et al. (2020), which are shown as dashed horizontal lines. In future scenarios, the GWL exceedance year occurs sooner in higher concentration scenarios than in lower concentrations scenarios. In all scenarios, the total anthropogenic carbon rises until at least the year 2050. In the two SSP1 scenarios, the total carbon starts to fall after this point, while it continues to grow in the other projections.

The fraction of carbon that is absorbed by the combined land and ocean reservoirs rises in the two SSP1 scenarios, remains approximately constant in SSP2-4.5 after 2050, and declines in the SSP3-7.0 and SSP5-8.5 scenarios. The time series at the year 2100 closely match the IPCC atmospheric fraction projections for the year 2100 (IPCC, 2021b, Fig. SPM.7), shown in

Anthropogenic Carbon Allocation Timeseries

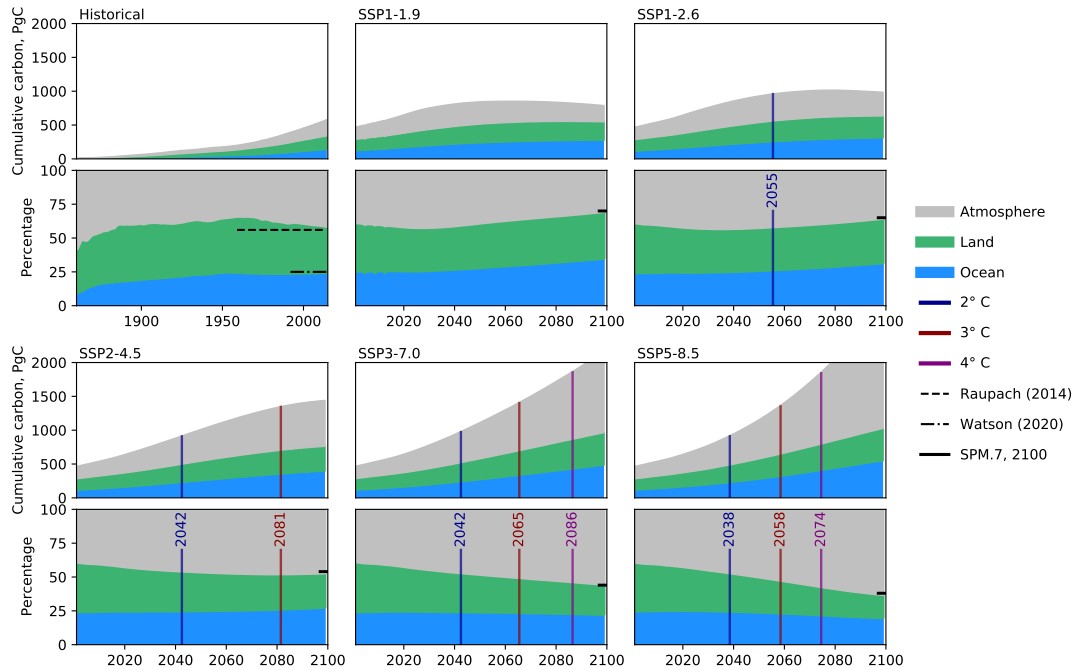

**Figure 3.** Multi-model mean carbon allocation time series for the historical period and each scenario. Each scenario includes a pair of panes: the top pane of each pair shows the total allocation in PgC, and the bottom pane shows the allocation as a percentage. The historical pane includes the observational records for the land and ocean fractions, from Raupach et al. (2014) & Watson et al. (2020), and the length of the lines represent the time over which the data was collected for these two observational datasets. The grey area is the cumulative anthropogenic carbon in the atmosphere, and the blue and green represent the fraction in the ocean and in the land, respectively. The SPM.7 lines at the year 2100 indicate the atmospheric fraction projections from the IPCC AR6 WG1 summary for policymakers figure 7, IPCC (2021b).

Fig. 3 as a short horizontal line at the end of the period. This corroboration with existing results increases confidence in the appropriateness of our methodology.

## 3.3 Multi-model ensemble carbon allocation

Figure 4 shows the carbon allocation at each GWL as a percentage (left) and in terms of the total carbon for each model (right). For each scenario and each GWL, the models are ordered by their ECS as shown in Table 1. The lower ECS models are at the top and higher ECS models are at the bottom of each section. The lower sensitivity models take longer to reach the same warming level and have more total emissions than the higher sensitivity models. This results in the saw-tooth pattern visible on the right of this figure.

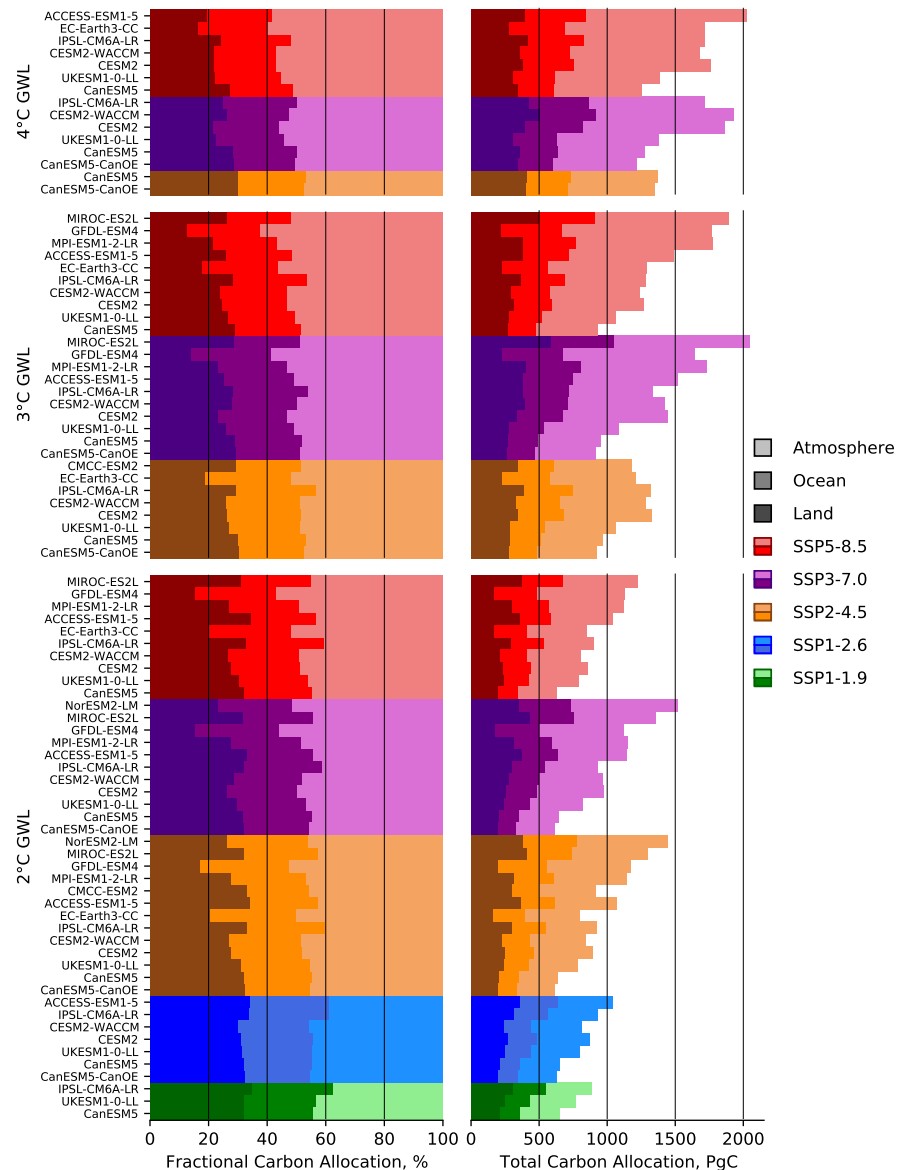

**Figure 4.** Global total carbon allocation for each level of warming for individual models. The left side shows the allocation as a percentage and the right side shows the total value in PgC. Each colour palette represents a different scenario, with SSP1-1.9 in greens, SSP1-2.6 in blues, SSP2-4.5 in oranges, SSP3-7.0 in purples and SSP5-8.5 in reds. The darkest shade denotes the land, the middle shade is the ocean and the lightest shade is the atmosphere. Within a given GWL and scenario, the models are ordered by their ECS, with less sensitive models at the top and more sensitive models at the bottom.

There is a significant variability between individual models in the total cumulative carbon allocated between scenarios at each GWL. For instance, the total carbon change at 2 °C ranges from 615 PgC (CanESM5-CanOE SSP3-7.0) to 1521 PgC

(NorESM2-LM SSP3-7.0). This range of behaviours between models is very large and the difference between these two extremes is equivalent to a century's worth of current global emissions, ie 100 years of $9.4 \pm 0.5$ PgC yr$^{-1}$ (Le Quéré et al., 2018).

Proportionally large ranges can also be seen in the land, ocean and atmospheric carbon sinks in Fig. 4. For instance, at the 2 °C GWL, the land has absorbed between 164 PgC (EC-Earth3-CC SSP2-4.5) and 432 PgC (MIROC-ES2L SSP3-7.0). Similarly, at the 2 °C GWL, the ocean has absorbed between 137 PgC (CanESM5-CanOE SSP3-7.0) and 401 PgC (NorESM2-LM SSP2-4.5). These ranges are equivalent to several decades worth of current global emissions, or approximately a century of the current annual rates of land or ocean carbon absorption. Almost all of the minimum and maximum values described here occur in the SSP3-7.0 scenario, for reasons discussed below in Sect. 4.2.

The left side of this figure shows several key results related to how carbon is allocated as a percentage of the total between models. At a given GWL, higher emission scenarios have a higher atmospheric fraction, a lower land fraction, and a relatively consistent ocean fraction. Warmer GWLs have larger atmospheric fractions, lower land fractions, and consistent ocean fractions than cooler GWLs.

## 3.4 Carbon allocation and ECS

The data from Fig. 4 is re-framed in Fig. 5 as a series of scatter plots. For each group of data, the line of best fit is calculated and the absolute value of the fitting error (Err, the standard error of the estimated gradient under the assumption of residual normality) divided by the slope (M) is shown in the legend as Err/M. This value indicates whether the slope crosses the origin within the 95% confidence limit (Err/M < 1) or not (Err/M > 1). While the value always appears in the legend, the line of best fit is only shown when Err/M < 1. All groups with three models or fewer that reach the GWL were excluded as there were not enough data points to draw meaningful conclusions.

We see that the GWL exceedance year, total carbon change and the individual total carbon allocation fractions are inversely correlated with ECS in Fig. 5. The GWL exceedance year and the total carbon allocations both have all absolute Err/M values lower than unity and are both related to ECS. The total carbon change is linked to ECS in both the ocean and the atmosphere, as their Err/M values are smaller than 1 in both cases. However, the correlations between carbon allocation fraction of the ocean and the atmosphere with ECS do not show a consistent correlation with ECS at any GWLs. For land, both the total carbon sink and the allocation fraction are not consistently correlated to ECS at all GWLs.

In addition to the GWL data, the values for the target year 2100 are shown in Fig. 5. The Err/M for the target year 2100 is greater than unity for the total carbon, the atmospheric carbon fraction, and both land columns, indicating a poor fit to a straight line. This indicates that ECS is not correlated to these quantities in the analysis around target years. Elsewhere, when the Err/M of the target year 2100 is less than unity, it is often close to unity or larger than the Err/M of the fits to the GWL data. This indicates that ECS is often less correlated to these quantities in target year analysis than they are in the GWL analysis. The GWL method allows us to characterise the impact of ECS, while the target year analysis obscures its influence.

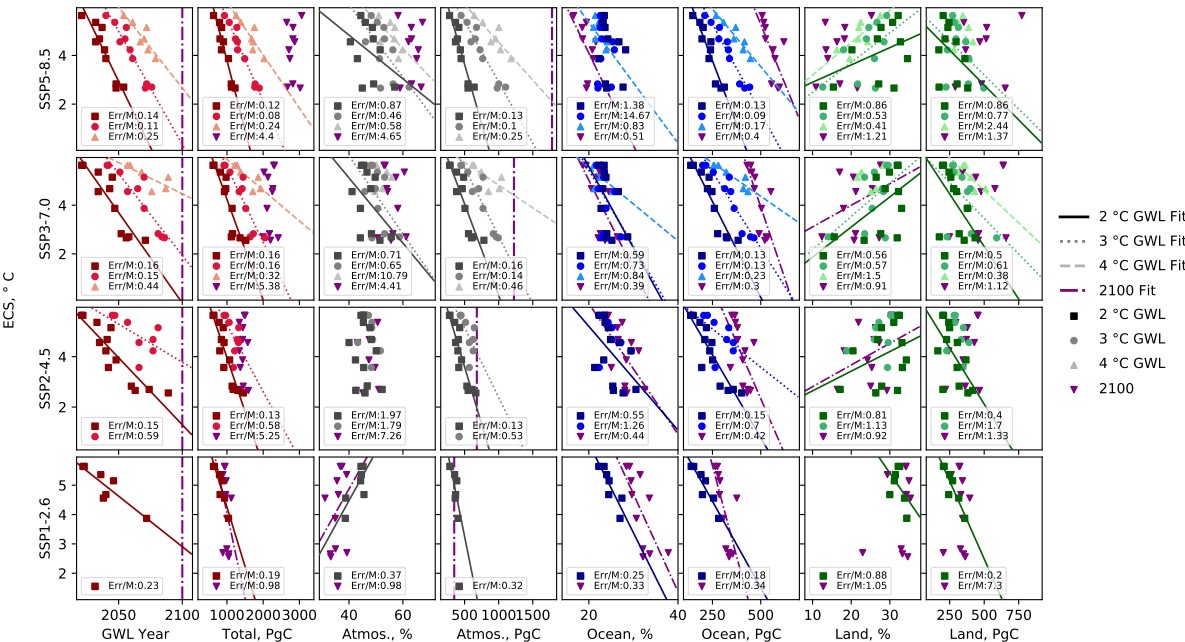

**Figure 5.** The GWL and target year 2100 carbon allocation scatter plot matrix. Each row represents a different scenario, and each column is a different data field, including the year, the total carbon allocated, the carbon allocation for each domain and the fractional carbon allocation to each domain. The y-axis is the model's ECS, and each point is a different GWL, where the squares are the $2°$ GWL, the circles are the $3°$ GWL, and the triangles are the $4°$ GWL. In all cases, the darkest colours correspond to the $2°$ GWL, the middle colours the $3°$ GWL, and the lightest colours the $4°$ GWL. The results for the target year 2100 are shown as purple downward-pointing triangles. The absolute value of the fitting error of the slope over the slope is shown in the legend as Err/M. The line of best fit is shown when Err/M $< 1$. The year 2100 and the total atmospheric carbon are indicated with purple vertical dash-dot lines.

## 4  Discussion

We present an analysis of the carbon allocation in the Earth System for an ensemble of CMIP6 simulations at the 2, 3 and 4 °C global warming levels. As described in in Sect. 1.3, using the GWL method instead of focusing on a specific target year allows us to provide estimates of the behaviour of the carbon cycle that may be more useful and relevant to policymakers. The difference between a focus on a specific target year and the GWL method can clearly be seen in Fig. 2 by comparing the top pane against the other three panes. At the year 2100, there are large differences across the five scenarios in the total carbon change, the allocation between the three reservoirs and the fractional distributions. In the lower three panes, the differences between scenarios is much smaller. However, these small differences are still significant in absolute terms, where several year's worth of global $CO_2$ emissions separate the scenarios at each GWL. The pathway to a given GWL is scenario-dependent in two main ways. Firstly, the rate of anthropogenic $CO_2$ emissions has a non-negligible impact on the atmospheric fraction because the ocean and land surface cannot quickly absorb the additional carbon load. A higher rate of emission leads to a slightly greater

transient warming, because fractionally more of the emitted $CO_2$ is still in the atmosphere. Secondly, $CO_2$ is the primary but not the only driver of warming. Differences in the non-$CO_2$ forcings for each scenario play a role in the realised warming at a given point in time. In addition, while the composition of each scenario ensemble results in a relatively uniform set of values of the mean ECS in Table 1, the mean ECS varies by up to 0.21 °C between scenarios. This could also account for some of the differences seen between multi-model means in Fig. 2. Furthermore, the SSP1-1.9 ensemble has the lowest mean ECS and the SSP5-8.5 ensemble has the highest mean ECS, which may exaggerate the differences between their multi-model means.

The GWL methodology allows a focused analysis on the small and subtle differences between scenarios. For instance in Canadell et al. (2021), Fig. 5.31 shows the cumulative carbon emissions against global mean temperature change for several projections. In that figure, all five projections show a strong correlation between $CO_2$ emissions and warming, all projections overlap at the same cumulative $CO_2$ emissions and there are no clear differences between scenarios for the same cumulative $CO_2$. Using the GWL method, we are able to focus on the differences between scenarios at the same warming level and demonstrate that small differences exist between scenarios and that the pathway to a GWL has an effect on the carbon allocation. While these differences in carbon allocation are highlighted by the GWL analysis, the differences between scenarios are consistent with previous studies and are likely due to differences in non-$CO_2$ forcing. However, it is beyond the scope of this work to quantify the non-$CO_2$ effect as Smith et al. (2020) have done.

On the left side of Fig. 2, the fraction of carbon that remains in the atmosphere is linked to the choice of scenario. The higher emission scenarios have higher atmospheric fractions at the same warming level. This is likely due to scenarios with higher carbon concentrations reaching the global warming levels sooner, and with proportionally less carbon allocated to the ocean and land surface at that time. In such cases, the ocean and the land have not caught up with the emissions or the warming associated with that $CO_2$ concentration. This implies that the carbon allocation between the three major sinks is likely impacted by the rate of warming at the GWL and there may be some delay between $CO_2$ emissions and the $CO_2$ atmospheric fraction reaching an equilibrium value since the excess $CO_2$ is absorbed more slowly by the terrestrial and oceanic sinks.

In the land surface at the 4 °C GWL, the multi-model mean land vegetation carbon increases by 384 and 436 PgC relative to 1850 in SSP5-8.5 and SSP3-7.0 respectively, as shown in Fig. 2. In CMIP5, the range relative to the years 1971-1999 was 52–477 PgC with a mean of 224 PgC, and was attributed mainly to $CO_2$ fertilisation of photosynthesis (Friend et al., 2014). While our CMIP6 multi-model mean is compatible with Friend et al. (2014), we do not see any individual model with only 52 PgC carbon allocated to the land at the 4 ° C GWL in Fig. 4. This absence is more likely to be attributed to the difference in the anomaly period (1850 vs 1971), rather than due to the significant changes between CMIP5 and CMIP6 land surface models. The land component model which contributed the value of 52 PgC to the CMIP5 analysis of Friend et al. (2014) was VISIT, which is part of the MIROC-ES2L ESM in CMIP6 (Hajima et al., 2020). However, MIROC-ES2L did not reach the 4 °C GWL in any scenario presented here. In all aspects of this analysis, the land carbon allocation has a much wider range of variability than the ocean. This reflects the significant challenge and uncertainty inherent in modelling the land surface carbon cycle (Friend et al., 2014; Jiang et al., 2019).

When comparing the same model at the same GWL across scenarios, the differences between scenarios becomes even more apparent, as shown in Fig. 4. This is especially true for low ECS models. For instance, the minimum and maximum carbon

allocation in MIROC-ES2L at 2 °C GWL is 1225 PgC in SSP5-8.5 and 1361 PgC in SSP3-7.0. The difference between these two projections of the same model with the same warming level is 136 PgC. For the decade 2008–2017, the mean annual emissions were $9.4 \pm 0.5$ PgC yr$^{-1}$, so this difference alone is approximately equivalent to 14 years of the current total global emissions.

The ocean maintains similar allocation percentages across the GWLs, but in Fig. 3 there is a small decline in ocean carbon allocation percentage at the highest $CO_2$ concentration scenarios towards the end of the 21$^{st}$ century. This is likely because much of the ocean is forecast to become increasingly stratified in the coming century, which would reduce downwards mixing of $CO_2$ (Li et al., 2020; Muilwijk et al., 2023). This reduction in downward mixing combined with the decline in solubility with rising sea surface temperature causes the overall absorption rate of $CO_2$ into the ocean to be reduced. The increase in

stratification is caused by warmer surface layers combined with gradual decline in overturning rates and overall circulation (Thibodeau et al., 2018; Li et al., 2020; Caesar et al., 2021; Sallée et al., 2021). Ocean acidification may also be playing a role in reducing the rate of the chemical transition of dissolved $CO_2$, thereby slowing uptake (Zeebe, 2012). In combination, these effects act to reduce the rate at which absorbed $CO_2$ is removed from the surface layer. In the ocean, enhanced ocean acidification has a range of effects but has been shown to decrease survival, calcification, growth, development and abundance

for a broad range of marine organisms (Kroeker et al., 2013).

    While the ocean fraction is more or less consistent throughout the SSP2-4.5, SSP3-7.0 and SSP5-8.5 scenarios at the GWLs, Fig. 3 shows that the land fraction declines over the coming century, from 35% at the end of the historical period to 25% in SSP2-4.5, 22% in SSP3-7.0 and 17% in SSP5-8.5 at the year 2100. The land fraction is forecast to decline over the coming century in the higher $CO_2$ concentration scenarios, although the total land carbon allocation increases. There are several pos-

sible explanations for this slowdown of uptake. Land ecosystems have been shown to become progressively less efficient at absorbing carbon as levels of atmospheric $CO_2$ concentrations increase (Wang et al., 2020). Some reasons for this could be that the soil respiration could increase due to warming more than any carbon uptake increases due to photosynthesis (Nyberg and Hovenden, 2020), or alternatively the nitrogen limitation could progressively limit photosynthetic uptake (Ågren et al., 2012). The changing climate may impact vegetation growth and photosynthetic uptake via droughts and warming, which changes the

temperature of plant-growing regions, thus decreasing the efficiency of photosynthesis. However, it is still not clear which of these factors have the largest impact.

    The differences in carbon allocations seen here have consequences for the Earth's climate. For example, global warming and higher $CO_2$ increases the regional and temporal variability of precipitation (Tebaldi et al., 2021). There is also the direct effect of increasing atmospheric $CO_2$ on radiative cooling rates. This impacts the vertical thermal structure of the atmosphere and

thus tropical overturning circulations and regional precipitation. This direct effect of atmospheric $CO_2$ is independent of the level of warming (Bony et al., 2013). This means that models or scenarios that have a greater atmospheric fraction of $CO_2$ at a given GWL will be more prone to this regional response to changed atmospheric radiative cooling, stability and circulation change, than models or scenarios with a smaller $CO_2$ fraction in the atmosphere.

## 4.1 Impact of ECS

The ensemble of CMIP6 models has a wide range of ECS values, and this impacts several aspects of carbon allocation. We have shown that the GWL exceedance year and the total carbon change are both inversely correlated with ECS. Similarly, we found that the carbon in the atmosphere and allocated to the ocean are both inversely correlated with ECS. The ECS does not appear to be consistently correlated with the total land carbon allocation or the land carbon fraction at any scenarios or GWL. The wider uncertainty and challenging nature of land surface carbon modelling is reflected in a broader range of behaviours in
land carbon models in CMIP6.

    In Fig. 4, when comparing individual models between different GWLs, the highest total carbon allocation at the 2 °C GWL is 1521 PgC in the SSP3-7.0 scenario in NorESM2-LM, which has an ECS of 2.56 °C. This is more carbon than several models emitted at higher GWLs: the lowest carbon emitted at 4 °C GWL was 1220 PgC for CanESM5-CanOE in the SSP3-7.0 scenario which has an ECS of 5.64 °C. In addition, both CanESM5 models (ECS: 5.64 °C) and the UKESM1-0-LL model
(ECS: 5.36 °C) reached 4 °C of warming in three different scenarios with less atmospheric carbon than NorESM2-LM had when it reached the 2 °C GWL. This highlights the significant role that ECS plays in the uncertainty of warming projections. A model's sensitivity to $CO_2$ concentration significantly impacts its projection of the total carbon allocation at global warming levels, as well as the absolute values of the individual carbon sinks in the ocean and land.

    The ECS impacts the GWL exceedance year, but this ensemble is also affected by survivor bias. While we hesitate to draw
conclusions from extrapolating the lines of best fit of Fig. 5, the line of best fit for the 2 °C GWL exceedance year for the SSP1-2.6 scenario crosses the year 2100 at an ECS equivalent to 3.1 °C. As the likely range of ECS values could be as low as 2.5 °C, this means that a non-trivial part of the ECS space could be excluded by the ScenarioMIP limit of forecasting to the year 2100. While we could extend the analysis with some longer term simulations, very few models and scenarios are available beyond the year 2100. To address this issue, the next round of ScenarioMIP in CMIP7 could extend its standard cutoff beyond
the year 2100. This would reduce survivor bias at 2 °C GWL and allow the inclusion of models having a low but still feasible ECS value of 2.5°C.

    Hausfather et al. (2022) outline a few analysis strategies for addressing the "hot model" problem in CMIP6. The first strategy is to use the GWL methodology as we have in this work. One of the alternative recommendations is to perform analysis of CMIP6 ensembles without the contributions of models that fall outside the likely ECS range of 2.5 - 4 °C. In our case, this would
remove seven of the thirteen models from the analysis, leaving six or fewer models contributing to each scenario. Based on our investigations in this paper, we believe that this would be an unnecessarily harsh requirement as we have already demonstrated that using GWL methodology can reduce the impact of the range of ECS relative to the "target year" methodology. In addition, uncertainties in cloud feedbacks have been identified as the main cause of the large range of ECS (Ceppi and Nowack, 2021), and it is unlikely that there is direct link between a model's ability to reproduce cloud feedback behaviour and its ability to
reproduce the carbon allocation, as these are independently modelled systems.

    We have used the terms equilibrium climate sensitivity (ECS) and effective climate sensitivity (EffCS) interchangeably. However, they are not the same. Gjermundsen et al. (2021) compared two Earth System models, NorESM2 and CESM2, that

had the same atmospheric model but different ocean components. These two models had very different EffCS values but were otherwise very similar: NorESM2's EffCS is 2.56 °C and CESM2's EffCS is 5.15 °C. In that work, they found that the greater heat storage at depth in NorESM2 delayed the Southern Ocean's surface warming and associated cloud responses, which in turn delayed the global mean surface warming. This effect appeared in the $4xCO_2$ simulation several centuries after the 150 year cutoff used to calculate climate sensitivity with the Gregory method (Gregory et al., 2004). After a sufficient number of simulated years, the same cloud feedback eventually occurs in both models, the same warming is realised, and the two models show similar equilibrium climate sensitivities. The Gregory method for calculating the effective climate sensitivity applied by Zelinka et al. (2020) to generate the ECS values used here does not capture all the details of the way a given cumulative emission produces a GWL, because it is not fully compatible with the true equilibrium climate sensitivity. It may be possible to take this effect into account in future works, for instance, by replacing the surface atmospheric warming anomaly with some measure of the global volume-weighted mean ocean heat anomaly.

## 4.2 Anomalous behaviour in SSP3-7.0

The SSP3-7.0 scenario is often an outlier throughout this analysis. For instance, in Figs. 2 and 4, it does not conform to the pattern of the other scenarios. In addition, SSP3-7.0 is the scenario showing the widest range of carbon allocation behaviours at both the 2 °C and 3 °C GWLs in Fig. 4. A possible reason for this is because the SSP3-7.0 scenario has the highest methane concentration and air pollution precursor emissions forcing, levels which are even higher than those of the SSP5-8.5 scenario (Meinshausen et al., 2017, 2020). In the other scenarios, the methane and aerosol precursors scale approximately in proportion to the $CO_2$ concentration. Methane is a strong greenhouse gas and has a warming effect, but pollution precursor emissions are linked to aerosols and cloud formation, which generally have a cooling effect (Twomey, 1977; Meinshausen et al., 2017). In CMIP6, methane warming can overwhelm, be overwhelmed by, or balance with aerosol cooling and the relative strengths of these effects depend strongly on the model parameterisation choices and their relative strengths in the scenario forcing. The relative strength of the warming methane emissions and the cooling aerosol precursors determines the impact on the warming rate and hence the GWL timing. This is why the warming in SSP3-7.0 is not as tightly bound to the atmospheric $CO_2$ concentration as in other scenarios. Even though warming is still correlated to total cumulative emissions, SSP3 scenarios may reach the GWLs relatively earlier or later than other scenarios at the same $CO_2$ concentration. This effect could be investigated in detail if for instance the SSP3-8.5 or SSP5-7.0 scenarios were simulated.

The impact of different methane and aerosol precursor emissions on the climate response remains highly uncertain in CMIP6 models. The overall warming impact of methane is not further considered in this work as is it secondary to $CO_2$ warming, but it could be examined in future extensions.

## 4.3 Limitations and possible extensions

While the CMIP6 experiments start their historical simulations in 1850 from a pre-industrial control, clearly this is not the starting point for the anthropogenic impact on the land surface or the carbon cycle. The human impact on the environment began much earlier and this has implications for on-going carbon partitioning (Bronselaer et al., 2017; Le Quéré et al., 2018;

Friedlingstein et al., 2022). For instance, between 1765 and 1850, atmospheric $CO_2$ rose by approximately 10 ppm, and accounting for this era resulted in a 4.5% change in ocean uptake in CMIP5 models (Bronselaer et al., 2017).

Similarly, the representation of dynamic vegetation, soil carbon and fire response is most likely under-sampled in this ensemble (Arora et al., 2020; Koch et al., 2021). Notably, CMIP6 models are not capturing present-day tropical forest carbon dynamics: the multi-model mean estimate of the pan-tropical carbon sink is half that of the observational estimate (Koch et al., 2021). These uncertainties in the strength of carbon-concentration and carbon-climate feedbacks over land are well-established (Cox et al., 2000; Friedlingstein et al., 2006; Arora et al., 2013).

The global ocean carbon inventory is also affected by the land-to-ocean carbon flux from river runoff and the carbon burial in ocean sediments, which is not represented in our ensemble (Arora et al., 2020). The flux of land carbon into the ocean via rivers is between $0.45 \pm 0.18$ PgC yr$^{-1}$ and $0.78 \pm 0.41$ PgC yr$^{-1}$ and is generally not considered in ESMs (Jacobson et al., 2007; Resplandy et al., 2018; Hauck et al., 2020). Including the riverine flux of particulate and dissolved organic carbon would require models to represent both estuarine and shallow shelf processes. This would most likely require higher model resolutions and computational costs.

One of the limitations of the GWL methodology is that it focuses on the realised warming at a specific point in time. This is the transient warming, and it is unlikely that this warming includes the full effect of all cumulative $CO_2$ emissions. In effect, the $CO_2$ emissions have not yet played out to equilibrium at the GWL, and there is likely to be a continued delay in their warming effect.

Not all scenarios are expected to reach these warming thresholds before the year 2100. For instance, while it is highly likely that all SSP5-8.5 simulations will reach 2 °C of warming, it is unlikely that any SSP1-1.9 simulations will reach 4 °C of warming. On the other hand, only some of the models reach the threshold in certain combinations of scenario and GWL. For instance, three of the six considered SSP1-1.9 models do not reach the 2 °C GWL. These missing models would most likely reach the thresholds at some point after the year 2100 if allowed to run for enough additional years with positive net $CO_2$ emissions. Future studies could potentially include the long-timeline simulations beyond the year 2100. The method that we used to populate Fig. 2 took the multi-model mean first with all models contributing equally, then used that ensemble mean to calculate the GWL exceedance years. An alternative method could first calculate the GWL exceedance years for individual ensemble members, then take the mean of only those that reach the threshold. However, this alternative method would implicitly include survivor bias, causing the overall weighting and conclusions to be biased towards high ECS models.

In this work, we used concentration-driven scenarios instead of emission driven scenarios. Emission driven scenarios allow significantly more flexibility in the behaviour of the atmospheric carbon. In practice, this would add a third degree of freedom into the total carbon allocation calculation. Although a limited set of emission driven runs exist, it was found that there are actually very few differences in simulated temperature or atmospheric $CO_2$ concentration between concentration driven and emission driven scenarios (Lee et al., 2021, Sect. 4.3.1.1). In any case, several key datasets required for the calculation of the land use emissions (*LUE* in Eq. 1) were not available in the emission driven experiments at the time of writing.

In Fig. 3, the multi-model mean of both SSP1 scenarios shows signs of recovery and carbon drawdown, but no datasets in this analysis drop below the 2 °C GWL exceedance. In future studies, it would be interesting to examine the reversibility

of carbon allocation with negative emission forcing scenarios. More generally, extension simulations beyond 2100 would be valuable for studying how patterns of carbon allocation change as emissions decline below net-zero, when carbon emissions become outpaced by carbon sinks.

In Fig. 5, we generated a fit to each dataset against the ECS. This fit is built on the assumption that these behaviours are linear and that the straight line fit is a reasonable approximation of their behaviour. However, as can be seen in this figure, this is not true in all cases. Several of the datasets have non-linear behaviours with regards to ECS. It may be possible to expand upon this work and generate more complex fits to these datasets to estimate the behaviour of these models within the likely ECS range of 2.5 - 4 °C.

In this work, although we attempt to maximise the number of models, ScenarioMIP's flexible specifications mean that each scenario's ensemble is composed of a different set of models, as shown in Table 1. This diversity results in a different mean ECS for each scenario. We were fortunate that the range of the mean ECS values was only 0.21 °C, despite for instance SSP1-1.9 containing significantly fewer models that the other scenarios. A different set of models could conceivably result in a wider range of mean ECS values across scenarios, which would impact the warming rates at the same $CO_2$ concentrations, making interpretation more challenging and potentially introducing bias in the conclusions. In future investigations of CMIP multi-model means using the GWL methodology, the mean equilibrium climate sensitivity of each ensemble should be included alongside the analysis as two ensembles consisting of differing sets of models may not always be directly comparable.

## 5   Conclusions

Using an ensemble of CMIP6 model simulations, we have quantified how the carbon allocation across Earth System components differs across scenarios after warming to the same global mean surface temperature anomaly. Scenarios with higher carbon concentrations reach these global warming levels sooner, and have proportionally less carbon allocated to the ocean and land surface at that time than scenarios with lower emissions. The differences in estimated carbon emissions between scenarios vary even at the same GWL, and can be equivalent to several years' worth of global total emissions. While these results arise as a direct consequence of using the GWL methodology, our conclusions are nevertheless compatible with previous works and we do not claim to refute previous target year analyses.

A model's sensitivity to $CO_2$ concentration significantly affects its total carbon allocation between the atmosphere, ocean and land at all global warming levels. However, our CMIP6 ensemble contains many models that fall outside the likely ECS range of 2.5 - 4 °C. By using the GWL methodology, we can exploit the full CMIP6 ensemble and weight each model equally, without excluding the so-called "hot models". We did not find a consistent relationship between ECS and any of the fractional carbon allocations, although, we did demonstrate that ECS and total carbon allocation are related. Models with lower sensitivity to carbon reach the GWL with more carbon in the individual reservoirs and more carbon overall. This is because it takes low ECS models longer to reach the same warming level, allowing more time for carbon to accumulate in the Earth System.

In addition to the impacts of ECS and total atmospheric carbon concentration, the distinct characteristics of each scenario pathway also influence the carbon allocation. The SSP3-7.0 scenario includes both high methane-induced warming and high

pollution precursors cooling, and the strength of these effects are model specific and not directly related to ECS. These environmental forcers in SSP3-7.0 can generate a very different warming response, GWL exceedance year and carbon allocation compared to scenarios where $CO_2$, methane and pollution precursors all scale with historical values.

Ultimately, across all model simulations, a significant rise in global mean surface temperature is projected over the 21$^{st}$ century. This underscores the need for an accelerating transition to low carbon technologies to reduce the risk of the worst effects of climate change.

*Code and data availability.* This analysis was performed using ESMValTool; the software tools used in this manuscript are available via zenodo: 10.5281/zenodo.8335060. The main ESMValTool recipe is `recipe_gwt_time_series_CMIP6_2022_all.yml` in the `esmvaltool/recipes` directory, and the main diagnostic script is `diagnostic_gwt_timeseries.py` in the `esmvaltool/diag_scripts/ocean` directory. An up-to-date version of the base ESMValTool system is available on github: github.com/ESMValGroup which includes up to date code, documentation and tutorials. CMIP6 climate model data used in this paper was obtained from the ESGF node at the Centre for Environmental Data Analysis (CEDA), but is widely available elsewhere: https://esgf-node.llnl.gov/search/cmip6/

*Video supplement.* A video abstract for this paper is available here: https://www.youtube.com/watch?v=abtNeOmb5XA.

*Author contributions.* All authors contributed to the writing, discussion, initial outline, literature survey and editorial feedback of the manuscript. LdM led the work, performed the analyses and led the writing. RS, CGJ, LdM developed the GWL analysis methods. CDJ, SL, TQ contributed to the land surface carbon calculation. JW contributed to the extraction and curation of the model data. CGJ, JCB led the UKESM and TerraFIRMA projects and working groups where this work was carried out.

*Competing interests.* The authors are not aware of any competing interests.

*Acknowledgements.* LdM and DIK were supported by the UK Natural Environment Research Council through The UK Earth System Modelling Project (UKESM, grant no. NE/N017951/1). RS, RJP, TQ and RPA are funded by the UK National Centre for Earth Observation (NE/N018079/1). LdM, RS, JCB, RJP, CDJ, CGJ, AY were supported by the UK Natural Environment Research Council through the TerraFIRMA: Future Impacts, Risks and Mitigation Actions in a changing Earth System project, Grant reference NE/W004895/1. CGJ acknowledges funding from the NERC National Capability UKESM grant no. NE/N017978/1 and EU Horizon 2020 project CRESCENDO, grant number: 641816. CDJ, SL and JW were supported by the Joint UK BEIS/Defra Met Office Hadley Centre Climate Programme (GA01101). CDJ was supported by the European Union's Horizon 2020 research and innovation programme under Grant Agreement No 101003536 (ESM2025 - Earth System Models for the Future). The authors would like to acknowledge use of the CEDA JASMIN computing cluster and British Atmospheric Data Centre (BADC) data centres in this work. The authors thank the JASMIN and ESMValTool teams for their assis-

tance with our analysis. Finally, the authors are grateful to anonymous reviewer #1 and Dr. John Dunne for their patience and contributions towards the final manuscript.

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
