# Peer review of "Scenario Choice Impacts Carbon Allocation Projection at Global Warming Levels"

_EGUsphere, 2022_

## Referee Comment (RC1)

**General comments**:

This article explores the carbon allocation with different choice of scenarios, SSP1-1.9, SSP1-2.6, SSP2-4.5, SSP3-7.0 and SSP5-8.5, at three different global warming levels (2, 3 and 4 degree celsius). Authors comprehensively include a wide range of ESM outputs and design a quantitative analysis framework to calculate carbon fractions in different reservoirs. The current version of the manuscript matches the scope of ESD and the presentation of methodology is enough. However, the main finding from this manuscript is not clear to me. Authors also need to heavily revise their results and discussion sections to provide logical and robust analysis and cross validation and comparison to previous studies.

**Specific comments**:

I have the following major comments:

1. You have some discussions about the implication of your study on the future carbon management and relevant studies in the discussion section, which is good. But the same information in the introduction part is missing. It would be nice to see more introduction about how carbon allocation is important for relevant research. For example, 1) how the calculated parameter can be important to the next stage of model intercomparison, benchmarking and 2) if this parameter can be helpful to indicate the strategy of carbon management for the next stage.

2. Contents in Results and Discussion sections are stacked in a whole block and require more revisions to streamline your manuscript structure. Please summarize 2-3 subtitles and split your context and fill in these sub-sections.

3. Line 231: "In summary, fig. 3 shows that a model's sensitivity to CO2 concentration significantly affects the total carbon allocation between the atmosphere, ocean and land at global warming levels, but is less impactful on the percentage allocation……the scenario has a much larger impact on the percentage carbon allocation at a given warming level than the ECS." But as I found in fig. 3, the carbon allocation fraction after normalization (left pane) are quite similar to each other under different scenarios at least for GWLs at 2 and 3 degree celsius. To the opposite, certain models show very large discrepancy, e.g. EC-Earth3-CC compared to other models. Please explain how you get this conclusion?

4. My understanding is that the authors plan to use UKESM as one of the examples to help understand how different processes in ESMs can influence the calculated carbon fraction. But I only find qualitative speculation instead of quantitative analysis. For example, in Line 340, "The UKESM1's higher AF at the year 2100 is likely due to the model limiting carbon uptake more than the other models. This could be Nitrogen limitation in the land surface or could be due to the model's higher ECS and thus warmer temperatures at 2100 than the multi-model mean." I expect to see more analysis, figures or tables to list evidence and prove these statements. Otherwise, there's no need to specifically highlight the result from one model and these conclusions from this manuscript are not robust.

5. In the discussion section, the manuscript lacks enough cross-validation or comparison against other similar published studies. There are published studies discussing carbon storage, residence time and feedbacks in land and ocean components under different future scenarios. Just to name a few here:

Friend, A. D., Lucht, W., Rademacher, T. T., Keribin, R., Betts, R., Cadule, P., et al. (2014). Carbon residence time dominates uncertainty in terrestrial vegetation responses to future climate and atmospheric CO2. Proceedings of the National Academy of Sciences, 111(9), 3280–3285. https://doi.org/10.1073/pnas.1222477110

Jiang, L., Yan, Y., Hararuk, O., Mikle, N., Xia, J., Shi, Z., et al. (2015). Scale-Dependent Performance of CMIP5 Earth System Models in Simulating Terrestrial Vegetation Carbon. Journal of Climate, 28(13), 5217–5232. https://doi.org/10.1175/JCLI-D-14-00270.1

Katavouta, A., & Williams, R. G. (2021). Ocean carbon cycle feedback in CMIP6 models: contributions from different basins. Biogeosciences, 18(10), 3189–3218. https://doi.org/10.5194/bg-18-3189-2021

6. Your key findings are not properly highlighted. To improve this draft, authors need to conclude a more solid and informative key finding, for example, "choice of forecast scenario impacts the carbon allocation at the same global warming levels more than model's ECS/TCRE". At the same time, provide more qualitative analysis to prove your key findings.

**Technical corrections and minor comments**:
Line 25: "and the land surface via primary production". Here "primary production" can be replaced by "terrestrial carbon fixation".
Line 27: "known as carbon allocation". To avoid confusion with the "carbon allocation" widely used in terrestrial ecosystem modeling, I would suggest clarifying this point here, such as "known as carbon allocation in the Earth Systems (we simply use carbon allocation in the rest of the text)".
Line 92: "land use emissions" contains how many different components? This LUE calculation may not contain the feedback from the settings of different ensembles.
Line 125: "can gives" shall be "can give"
Line 131: "Individual component models can be used by" can be clarified as "Same Individual component model can be used by".
Line 137: Please clarify "All quoted values". What are these values?
Line 148: "In addition, several models may share contributing component models" seems to be a repetition of the content in Line 131. Shall think about how to merge them.
Line 165: "These tools include quick ways to standardise, slice, re-grid, and apply statistical operators to datasets." Can you provide a table or figure to summarize and explain the mathematical algorithms of the operators you applied in this paper through using ESMValTool for data pre-processing? I think this is necessary information to understand your methodology.
Line 193: "Figure 2 only shows the multi-model means, not single models." It will be helpful to add the spread of carbon allocation fraction using the results from single models in figure 2.

Line 302: "Therefore, SSP3-7.0 can reaches" shall be "reach".
Line 302: "Therefore, SSP3-7.0 can reaches the GWLs earlier than other scenarios at the same CO2 concentration". I'm not quite sure about this conclusion. If we take a look at figure 4, SSP3-7.0 is later than SSP5-8.5 to reach all 3 GWLs.
Line 315: "Higher CO2 is causes" shall be "Higher CO2 causes"
Line 323: "the rate at which surface waters and dissolved CO2 is mixed downward will slow. This reduction is downward mixing reduces the overall absorption rate of CO2 into the ocean" This statement is confusing. Please rephrase.

Figure 1: It's better to clarify that your prescribed DCO2 has accounted for the anthropogenic fossil fuel exploitation and the subsequent C emission from application.
Figure 4: "the historical observations from Raupach et al. (2014) & Watson et al. (2020)," It will be better if you can clarify in which year(s) these observations represent.

There are plenty of other typos and confusing statements in this draft and the authors shall be responsible to double check the whole document before resubmission.

---

## Author Comment (AC1)

**Reply to Referees comments - Choice of Forecast Scenario Impacts the Carbon Allocation at the Same Global Warming Levels**
Lee de Mora et al.

The authors would like to thank the editor, Somnath Baidya Roy, Anonymous Referee #1 and John Dunne for their efforts. Thank you all for taking the time to read the manuscript and share your comments. Your comments have been taken on board and the manuscript is in a much better place now after these revisions.

Both referees highlighted the need for a clearer articulation of the main findings. In order to clarify our findings, we have followed Anonymous Referee #1 suggestion and re-written significant parts of the paper, including the abstract, introduction, results, the discussion, conclusions. We have revised several figures, included a new figure that shows this result explicitly and added a new table of results numerically.

We are still in the process of updating the manuscript, but we have not found anything in the review that we would want to flag as a problem or a showstopper.

Below this introductory section is a reply to each of the comments point by point. Our responses are marked in *blue italics*. For the technical and minor revisions, we will have implemented the changes below, but not all of the original text survived into the revised manuscript.

General Comments:

This article explores the carbon allocation with different choice of scenarios, SSP1-1.9, SSP1-2.6, SSP2-4.5, SSP3-7.0 and SSP5-8.5, at three different global warming levels (2, 3 and 4 degree Celsius). Authors comprehensively include a wide range of ESM outputs and design a quantitative analysis framework to calculate carbon fractions in different reservoirs. The current version of the manuscript matches the scope of ESD and the presentation of methodology is enough.

*LdM: Thank you for a clear summary of the work. We're glad that it is within scope for ESD.*

However, the main finding from this manuscript is not clear to me. Authors also need to heavily revise their results and discussion sections to provide logical and robust analysis and cross validation and comparison to previous studies.

*LdM: We have revised the abstract, introduction, discussions and conclusions sections. Added a new figure to clear up our conclusions and removed the focus on UKESM as a standalone model. We hope that these changes are sufficient to address this comment.*

Specific comments:

I have the following major comments:

1. You have some discussions about the implication of your study on the future carbon management and relevant studies in the discussion section, which is good. But the same information in the introduction part is missing. It would be nice to see more introduction about how carbon allocation is important for relevant research. For example, 1) how the calculated parameter can be important to the next stage of model intercomparison, benchmarking and 2) if this parameter can be helpful to indicate the strategy of carbon management for the next stage.

*LdM: We have revised the introduction with a wider description of carbon allocation and recent research in this area.*

2. Contents in Results and Discussion sections are stacked in a whole block and require more revisions to streamline your manuscript structure. Please summarize 2-3 subtitles and split your context and fill in these sub-sections.

*LdM: We have revised both the result and discussion sections and added sub-headings.*

3. Line 231: "In summary, fig. 3 shows that a model's sensitivity to CO2 concentration significantly affects the total carbon allocation between the

atmosphere, ocean and land at global warming levels, but is less impactful on the percentage allocation......the scenario has a much larger impact on the percentage carbon allocation at a given warming level than the ECS." But as I found in fig. 3, the carbon allocation fraction after normalization (left pane) are quite similar to each other under different scenarios at least for GWLs at 2 and 3 degree Celsius. To the opposite, certain models show very large discrepancy, e.g. EC-Earth3-CC compared to other models. Please explain how you get this conclusion?

*LdM: We have added a new figure to the results section to help clarify this conclusion. We have also re-written this section to be more clear.*

4. My understanding is that the authors plan to use UKESM as one of the examples to help understand how different processes in ESMs can influence the calculated carbon fraction. But I only find qualitative speculation instead of quantitative analysis. For example, in Line 340, "The UKESM1's higher AF at the year 2100 is likely due to the model limiting carbon uptake more than the other models. This could be Nitrogen limitation in the land surface or could be due to the model's higher ECS and thus warmer temperatures at 2100 than the multi-model mean." I expect to see more analysis, figures or tables to list evidence and prove these statements.  Otherwise, there's no need to specifically highlight the result from one model and these conclusions from this manuscript are not robust.

*LdM: We have removed the focus on the UKESM model now.*

5. In the discussion section, the manuscript lacks enough cross-validation or comparison against other similar published studies. There are published studies discussing carbon storage, residence time and feedbacks in land and ocean components under different future scenarios. Just to name a few here:

Friend, A. D., Lucht, W., Rademacher, T. T., Keribin, R., Betts, R., Cadule, P., et al. (2014). Carbon residence time dominates uncertainty in terrestrial vegetation responses to future climate and atmospheric CO2. Proceedings of the National Academy of Sciences, 111(9), 3280–3285. https://doi.org/10.1073/pnas.1222477110

Jiang, L., Yan, Y., Hararuk, O., Mikle, N., Xia, J., Shi, Z., et al. (2015). Scale-Dependent Performance of CMIP5 Earth System Models in Simulating Terrestrial Vegetation Carbon. Journal of Climate, 28(13), 5217–5232. https://doi.org/10.1175/JCLI-D-14-00270.1

Katavouta, A., & Williams, R. G. (2021). Ocean carbon cycle feedback in CMIP6 models: contributions from different basins. Biogeosciences, 18(10), 3189–3218. https://doi.org/10.5194/bg-18-3189-2021

*LdM: Thanks for these references, I particularly liked the Katavouta one, what a great paper! We have added used them to clarify our results and add some cross-sectional validation. We've also added a paragraph comparing our results to Friends 2013 work.*

6. Your key findings are not properly highlighted. To improve this draft, authors need to conclude a more solid and informative key finding, for example, "choice of forecast scenario impacts the carbon allocation at the same global warming levels more than model's ECS/TCRE". At the same time, provide more qualitative analysis to prove your key findings.

*LdM: We have added a new figure to the manuscript to clarify our conclusions with regards to the carbon allocation.*

**Technical corrections and minor comments**:

Line 25: "and the land surface via primary production". Here "primary production" can be replaced by "terrestrial carbon fixation".

*LdM: Done*

Line 27: "known as carbon allocation". To avoid confusion with the "carbon allocation" widely used in terrestrial ecosystem modeling, I would suggest clarifying this point here, such as "known as carbon allocation in the Earth Systems (we simply use carbon allocation in the rest of the text)".

*LdM: Done*

Line 92: "land use emissions" contains how many different components? This LUE calculation may not contain the feedback from the settings of different ensembles.

*LdM: Added more details on how LUE was calculated in Liddicot 2021*

Line 125: "can gives" shall be "can give"

*LdM: Done*

Line 131: "Individual component models can be used by" can be clarified as "Same Individual component model can be used by".

*LdM: Done*

Line 137: Please clarify "All quoted values". What are these values?

*LdM: Clarification added to text*

Line 148: "In addition, several models may share contributing component models" seems to be a repetition of the content in Line 131. Shall think about how to merge them.

*LdM: Fixed this.*

Line 165: "These tools include quick ways to standardise, slice, re-grid, and apply statistical operators to datasets." Can you provide a table or figure to summarize and explain the mathematical algorithms of the operators you applied in this paper through using ESMValTool for data pre-processing? I think this is necessary information to understand your methodology.

*LdM: We share the ESMValTool recipe and the source code, but the actual method is relatively simple. I've added the sentence to be explicit about our calculation:*

*In our case, we used the \textsc{annual\_statistics} preprocessor to calculate the annual mean, the \textsc{mask\_landsea} preprocessor to mask the land or sea areas, and the \textsc{area\_statistics} preprocessor to calculate the area weighted global mean.*

Line 193: "Figure 2 only shows the multi-model means, not single models." It will be helpful to add the spread of carbon allocation fraction using the results from single models in figure 2.

*LdM: The spread of individual models is shown in figure 3.*

Line 302: "Therefore, SSP3-7.0 can reaches" shall be "reach".

*LdM: fixed*

Line 302: "Therefore, SSP3-7.0 can reaches the GWLs earlier than other scenarios at the same CO2 concentration". I'm not quite sure about this conclusion. If we take a look at figure 4, SSP3-7.0 is later than SSP5-8.5 to reach all 3 GWLs.

*LdM: I think we may have mis-communicated this result. This entire paragraph has been re-written for clarity.*

Line 315: "Higher CO2 is causes" shall be "Higher CO2 causes"

*LdM: fixed*

Line 323: "the rate at which surface waters and dissolved CO2 is mixed downward will slow. This reduction is downward mixing reduces the overall absorption rate of CO2 into the ocean" This statement is confusing. Please rephrase.

*LdM: This has been rephrased to:*
*This is likely because the surface layers of the ocean will be in equilibrium with the atmosphere, while deeper layers are not. However, much of the ocean is forecast to become increasingly stratified in the coming century, which would reduce downwards mixing of $CO_2$.*

Figure 1: It's better to clarify that your prescribed DCO2 has accounted for the anthropogenic fossil fuel exploitation and the subsequent C emission from application.

*LdM: We added this to the figure caption.*

Figure 4: "the historical observations from Raupach et al. (2014) & Watson et al. (2020)," It will be better if you can clarify in which year(s) these observations represent.

*LdM: The length of the lines represent the time over which the data was collected for these two observational datasets.*

There are plenty of other **typos and confusing state**ments in this draft and the authors shall be responsible to double check the whole document before resubmission.

*LdM: We can only apologise for these unfortunate problems. We have been more cautious with the revised manuscript.*

Reply

**Citation**: https://doi.org/10.5194/egusphere-2022-1483-RC1

---

## Author Comment (AC2)

**Reply to Referees comments - Choice of Forecast Scenario Impacts the Carbon Allocation at the Same Global Warming Levels**
Lee de Mora et al.

The authors would like to thank the editor, Somnath Baidya Roy, Anonymous Referee #1 and John Dunne for their efforts. Thank you all for taking the time to read the manuscript and share your comments. Your comments have been taken on board and the manuscript is in a much better place now after these revisions.

Both referees highlighted the need for a clearer articulation of the main findings. In order to clarify our findings, we have followed Anonymous Referee #1 suggestion and re-written significant parts of the paper, including the abstract, introduction, results, the discussion, conclusions. We have revised several figures, included a new figure that shows this result explicitly and added a new table of results numerically.

We are still in the process of updating the manuscript, but we have not found anything in the review that we would want to flag as a problem or a showstopper.

Below this introductory section is a reply to each of the comments point by point. Our responses are marked in *blue italics*. For the technical and minor revisions, we will have implemented the changes below, but not all of the original text survived into the revised manuscript.

RC2: 'Comment on egusphere-2022-1483', John Dunne, 08 Feb 2023

The manuscript "Choice of Forecast Scenario Impacts the Carbon Allocation at the Same Global Warming Levels" by de Mora et al provides an analysis of the carbon allocation across land, atmosphere, and ocean across a subset of CMIP6 models. While I was somewhat surprised at the degree of model agreement, The analysis and conclusions are fairly straightforward and of value to the broad audience of carbon cycle researchers.

*LdM: Thanks for the summary and kind words!*

I have detailed many specific examples of technical questions and points of clarification that I thought should be addressed before publication. It would also be helpful to add more information on caveats that might lead to an underestimation of the overall uncertainty. For example, while the CMIP6 historical simulations start in 1850, it is understood that changes to the carbon cycle began well beforehand which has implications for ongoing partitioning (Bronselaer et al., 2017 https://agupubs.onlinelibrary.wiley.com/doi/10.1002/2017GL074435 ; Le Quere et al. 2018 https://essd.copernicus.org/articles/10/2141/2018/).

Similarly, representation of dynamic vegetation, soil carbon and fire response is most likely undersampled in this ensemble (Arora et al., 2020 https://bg.copernicus.org/articles/17/4173/2020/bg-17-4173-2020.pdf ; Koch et al., 2021 https://agupubs.onlinelibrary.wiley.com/doi/full/10.1029/2020EF001874 ).

*LdM: This is a great point, thanks for pointing us towards these interesting results! We have added this discussion to the limitations section of the manuscript.*

Specific comments:
Title – "forecast", which implies an initial value problem is inappropriate and should be "projection" which implies a boundary value problem.

*LdM: Changed forecast to projection in the title*

Abstract, line 13 – Albeit not having read the rest of the manuscript at this point, after hearing that the range of carbon allocation between scenarios towards 2C varies by only 3%, I find the conclusion, "However, the choice of scenario has a much larger impact on the percentage carbon allocation at a given warming level than the individual model's ECS". Difficult to understand/believe**…**are the authors only referring the ECS as an indicator of the differing model approach to 2C, or to the overall ECS over CO2 doubling, which might vary from 2-5C or more?  I believe the authors are only referring to the pace of attaining 2C which is far more specific than the current statement conveys.  For example, approaching the equilibrium temperature at CO2 doubling or even 3C could have very different implications for carbon allocation than the scenario approach to 2C. (Note, upon finishing the manuscript, I felt like this issue was not resolved).

*LdM: We have re-written the abstract, introduction and other parts of the manuscript in order to be explicitly clear in our results.*

– "(" belongs before "Ukkola"

*LdM: Fixed*

– "that we have" is unnecessary

*LdM: Fixed*

– add comma after "fuels"

*LdM: done*

– "tool that we have to make forecasts of the future climate" should be "tools capable of projecting the future coupled carbon-climate system"

*LdM: fixed*

– "This means that the model outputs must use a common format and meet the minimum quality requirements." Adds nothing beyond the previous sentence

*LdM: removed*

– "…drift in the global volume mean ocean temperature of less than 0.1 degrees per year."  Are you sure about this?  A mean ocean temperature change of 0.1 C per year corresponds to a global radiative imbalance at the ocean surface of about 60 W per m2… about 100 times greater than the present day imbalance… are you sure that isn't supposed to be "0.1 degrees per century"?

*LdM: Yes, indeed it should be per century.*

– "forecast" should be "scenario"

*LdM: fixed*

– "breaks" should be "break"

*LdM: fixed*

– comma after "year"

*LdM: fixed*

– While the statement "and several members of the authorship team contributed to the development of the UKESM1 model" may be relevant to the execution of the manuscript and important to establish author contributions, it is not appropriate to provide in the manuscript content.

*LdM: Removed this.*

– The sentence "This is typically expressed as an annual total, so the total cumulative flux is calculated as the cumulative sum of the global annual total fluxes along the time dimension" is redundant in invoking "total" 3 times, and "annual" and cumulative" twice.

*LdM: Simplified this sentence for clarity.*

– The statement "Here, we take land-use emissions from the scenario, so they are not in balance with run-time model behaviour: this means that SLAND is only an approximation." Is unclear as to the need for an approximation. More information on how land use fluxes are treated is warranted. Why is a precise budget not possible? How much uncertainty is there in this "approximation"?

*LdM: Sorry for the confusion, it is not that land use is not included in the runs, but that the impact of land-use on carbon stores is not able to be diagnosed. This is because changes in land carbon include natural and human-caused. Therefore we can't estimate total emissions from each model, only the fossil fuel component. This is standard – e.g. see figure in Jones et al 2013 (https://journals.ametsoc.org/view/journals/clim/26/13/jcli-d-12-00554.1.xml, fig A1), and this approach also taken in IPCC AR6 (as explained in caption for figure SPM.7).*

– "may appear in several of the earth system models"... The word "may" here is inappropriate.

*LdM: we have changed our language to be more precise here.*

- In which of the models used in the present study is the same version of the NEMO circulation model used? This should be specific. How does the model diversity sampled here, in weighting the NEMO model. impact the overall diversity captured in the larger ensemble in CMIP5 and CMIP6, for example, including the GFDL results in the idealized experiments as was done in Arora et al., 2020 (https://bg.copernicus.org/articles/17/4173/2020/bg-17-4173-2020.pdf)

*LdM: We do not want to include a table like Arora, as this information is widely available elsewhere. However, we have changed the text:*

*In addition, the same individual component models are used by several modelling centres. For instance, the NEMO ocean circulation model forms the marine circulation component model of six of the earth system models used here \citep{Heuze2021}.*

– The word "weighted" is inappropriately vague here, since the "one-model one-vote" approach was used. The word should be "mean", or "median" as appropriate.

*LdM: changed to "This table also shows the mean ECS of the contributing models for each scenario."*

Table 1 – Why wasn't the GFDL-ESM4 model included? It has among the most sophisticated treatments of vegetation/land use and ocean biogeochemistry and is the highest performer in reproducing historical warming (Brunner et al., 2020; https://esd.copernicus.org/articles/11/995/2020/).

*LdM: GFDL-ESM4 models was absent because our code excluded it. The reason is that it uses a non-standard grid label (gr1) in CMIP6 Amon and Lmon, so our tools didn't find it. We've added it into the ensemble in the revised draft, but it's presence doesn't change the overall conclusions. (GFDL-CM4 data remains excluded because it does not provide the nbp field required for the land component of the analysis.)*

- What is the support for "These model pairs are likely only to have slight differences."? Similar to the assertion that multiply models use the same ocean, these characteristics should be justified. There are many previous intermodal comparisons on "uniqueness" and "independence" including the Brunner paper mentioned above that could be referenced on this.

*LdM: This statement was unjustified and has been removed.*

178, 183 – Should "SSP1-2.5" be "SSP1-2.6"?

*LdM: Fixed*

– I don't know what is being referred to as "This is known as survivor bias". What is "This" The lack of some models to meet a metric?

*LdM: Change this to: "If we were to draw conclusions uniquely using models that reach this threshold, then those conclusions would be influenced by survivor bias."*

– What do the authors mean by "strange behavior"?

*LdM: Changed this to: "This model also exhibited outlier behaviour in CMIP5 (Dunning 2018)"*

– The phrase "and if the atmospheric carbon concentration were allowed to rise sufficiently high" is not a necessary condition for warming based on TCRE – as long as emissions are positive, temperatures are expected to rise even if concentrations are declining.  The statement should rather be "and if net CO2 emissions are positive"

*LdM: Good spot! Changed it to:  "if the model were allowed to run for long enough with positive net CO2 emissions."*

– The assertion that ocean variability is larger than land variability in "The variability in the ocean is likely due to the wider range of circulation behavior in the scenarios." Seems very difficult to believe given the dominant role of land variability in historical interannual variability in carbon uptake as documented by the Global Carbon Project and IPCC… is this an indication of a lack of realism in the UKESM1 representation of interannual carbon variability on land, either through lack of ENSO variability or the land response?  Perhaps I don't understand well enough how this is being calculated to average out land carbon internal variability, or if the models chosen do not have reasonable amount of historical variability.  More explanation is warranted.

*LdM: We have removed the focus on the UKESM section of the manuscript.*

– comma after "land"

*LdM: Done*

– The end of the sentence is confusing to me as I do not understand how some models achieve "similar atmospheric CO2 concentrations" with "faster atmospheric CO2 growth" than others… "This means that even though two scenarios may reach the same warming level with similar atmospheric CO2 concentrations, the ocean and the land surface absorb less carbon in the scenario with faster atmospheric CO2 growth." Are the authors saying that the same GWL can be achieved at the same atmospheric CO2 concentration by both a high ECS model early in SSP585 as well as a low ECS model in SSP245?  Some explanation and examples are necessary.

*LdM: We have removed this explanation and added an ECS correlation testing exercise to help unpick some of these behaviours.*

– Given that representation of methane and aerosol precursor emissions have been studied for decades and played a major role in both CMIP5 and CMIP6 (much of the focus of AR6 WGI Ch6), I do not think the word "infancy is accurate in the sentence "The impact of different methane and aerosol precursor emissions on the climate response is still in its infancy in terms of realism in CMIP6." Rather I think it would be more accurate to stay that these topics remain highly uncertain.

*LdM: Fixed.*

– move "(" to before "Wang"

*LdM: fixed*

– remove "is"

*LdM: fixed*

– "reduction is" should be "reduction in"

*LdM: fixed*

– The logic here is reversed – "more saline surface layers" decreases stratification rather than increasing it.

*LdM: We removed this.*

– move "(" to before "Zeebe", also, remove "together"

*LdM: done*

– remove "which"

*LdM: done*

– remove second "could be"

*LdM: done*

Reply
**Citation**: https://doi.org/10.5194/egusphere-2022-1483-RC2

---

## Author Response (AR2)

**Scenario Choice Impacts Carbon Allocation Projection at Global Warming Levels - Authors response to reviewers**

Hi,

Thanks to both reviewers and the editor for their on-going work with this manuscript. We appreciate you taking the time to review the paper a second time and the changes that you've suggested made have improved the manuscript.

First, we went through the comments and addressed every comment. Then we did a full read-through and edit. This means that some of the comments, particularly grammar and typos were fixed, but then moved elsewhere in the text, removed entirely or re-written.

We have responded to each point below in ***bold italics.***

Kind regards,

Lee de Mora and the Authorship team.

The manuscript "Scenario Choice Impacts Carbon Allocation Projection at Global Warming Levels" by de Mora et al provides a useful analysis framework for interpreting carbon emissions allocation between atmosphere, land and ocean across the Shared Socioeconomic Pathway (SSP) future projections out to 2100 used in the Sixth phase of the Coupled Model Intercomparison Project, quantifying the tendency of more carbon remaining in the atmosphere under scenarios of faster emissions to achieve the same Global Warming Level (GWL).

*LdM: Thanks for the kind words, John.*

What I missed was the value of this approach in achieving consistency among carbon cycle responses relative to what the authors point as the more common approach of reference time frames (2050, or 2100). For example, how much larger would the variance in Figure 4 have been if the models were grouped by time rather than warming level for each scenario? The answer to this question would seem a critical justification for this approach in being able to simplify the interpretation of the carbon cycle response– in the same way that the Transient Cumulative Response to Emissions (TCRE) concept helped simplify communication of the coupled carbon-climate response. Currently, the main justification appears to salvage information from the models with unrealistically high ECS, which is much less compelling.

*LdM: Apologies for the long answer, but there are a few questions here. Firstly, the question of including an analysis of specific points in time as well as the GWL. Secondly, the question about "salvaging information from high ECS models", which we interpret as two separate questions:*

- *What is the value of GWL analysis?*
- *Why should we include high ECS models in our analysis?*

*We also attempt to address some of the points related to these questions raised elsewhere in the review here.*

*We do partially answer the question of grouping by time in the top pane of figure 2, which shows the multi-model mean at the year 2100. The following figure is a version of figure 4 with data from the target year 2100 only. In this figure, the variability between scenarios and models is in line with the results seen in the top pane of figure 2. Going from low emission scenarios to high emission scenarios, we see the expected behaviour, where the fraction of CO2 in the atmosphere rises.*

[Figure]

 We chose not to include this figure in the manuscript because we want to answer a specific question: "how does the world look after 2, 3 and 4 degrees of warming?" This approach has been applied to other climate quantities - such as R. Swaminathan's 2022 paper (https://doi.org/10.1175/JCLI-D-21-0234.1). There is a growing set of literature in this area as the GWL methodology is increasingly relevant for policymakers. In this work, we found that some parts of the carbon allocation have a path dependence, which is expected due to the timescales of the processes, but is nevertheless is a novel finding.

We have amended figure 5, also included below, such that it is shows the 2100 target year data, as well as data from the three GWLs. In this new version of the figure, the values of the target year 2100 are shown in as purple inverted triangles, and a line of best fit is shown as a purple dash-dot line when Err/M <1. For the year in the first column, the year 2100 is shown as a vertical line, and the total atmospherically carbon column has a vertical line for the atmospheric carbon at the year 2100. The Err/M for the target year 2100 is greater than unity in the total carbon, the atmospheric carbon fraction, and both land columns, indicating a poor fit to a straight line. This indicates that ECS is not correlated to these data in target year analysis.  Elsewhere when Err/M the target year 2100 is less than one, it is almost always larger than the Err/M of the fits to the GWL data. The GWL method allows us to characterise the impact of ECS, while the target year method obscures its influence.

[Figure]

The next part of the question was: "What is the value of GWL analysis?" The GWL methodology is an appropriate way to understand the projections of our models at specific policy-relevant warming levels and it is independent of the range of ECS values of the ensemble. This method existed before CMIP6's ECS range was known (James et al 2017 https://doi.org/10.1002/wcc.457). The advantages are well documented:

> "[The GWL methodology] mirrors the policy discourse surrounding the Paris agreement targets of 1.5 °C and 'well below 2 °C'. It is also largely independent of the choice of future emissions scenario — despite some differences related to the rate of warming and aerosol forcing, the world largely looks the same at 2 °C, no matter how we get there. And, to a certain extent, using global warming levels bypasses the need to select or weight CMIP6 models. Each model has something to offer at a given temperature, so the full CMIP6 ensemble can be used." Hausfather et al 2022  (https://www.nature.com/articles/d41586-022-01192-2)

The third part of the question was "Why should we include high ECS models in our analysis?" ECS is largely determined by a models ability to reproduce cloud-climate feedbacks. As such, ECS is only one aspect of model quality. There is no reason to expect the carbon cycle to be related to the model's ability to represent cloud feedbacks. In the same way that cloud modellers don't throw away models with (for example) unrealistic marine biogeochemistry, we choose not to restrict our carbon cycle analysis to those models that happen to present global-scale cloud feedbacks with the likely range.

The Hausfather et al (2022) comment referenced above is one of the sources of the idea that models should be ruled out simply on the basis of their ECS. However, even in that work, they first suggest using GWL methods before discarding models based on ECS or TRCE.  Some of our authorship team [CDJ] are currently preparing a rebuttal to that work and other rebuttals and other shorter comments already exist (ie https://www.nature.com/articles/d41586-022-02241-6). While this is not the place for a full rebuttal, the proposal to exclude models based on ECS or TCR is

*an over-simplistic approach which should be used sparingly, and would be scientifically inappropriate in our case. Once we start whitling down our ensemble by removing models outside the accepted range, we remove their contributions and potentially bias our results towards the assumptions that underlie the models with the ECS between 2.5 and 4 degree C.*

*In a pragmatic sense, this was a particularly challenging ensemble to build, as we required multiple simultaneous datasets, which reduces to overall size of the ensemble. We have only 13 models in the ensemble in total. The following figure shows figure 4 if we were to remove the 7 models that had ECS outside the likely range. The broad patterns we describe in our manuscript are still there (more or less), but the overall results are much less compelling.*

[Figure]

*In either case, we have summarised these arguments in the following paragraph in the Impact of ECS section of the discussion:*

> *Hausfather et al (2022) outline a few analysis strategies for addressing the "hot model" problem in CMIP6. The first option is to use the GWL methodology as we have in this work. One of the alternative recommendations is to perform analysis of CMIP6 ensembles without the contributions of models that fall outside the likely ECS range of 2.5 - 4 ◦C. In our case, this would remove seven of the thirteen models from the analysis, leaving six or fewer models contributing to each scenario. This would be an unnecessarily harsh requirement as we have already demonstrated that using GWL methodology can reduce the impact of the range of ECS relative to the "target year" methodology. In addition, uncertainties in cloud feedbacks have been identified as the main cause of the large range of ECS (Ceppi and Nowack, 2021), and it is unlikely that there is direct link between a models ability to reproduce cloud feedback behaviour and its ability to reproduce the carbon allocation, as these are independently modelled systems.*

The discussion was somewhat wandering, and I suggested some areas that could be cut if space is an issue. My big frustration with this manuscript was the number of grammar mistakes, particularly given the number of English Institutions and native English speakers represented. I eventually stopped pointing out grammar mistakes in my technical comments in interest of time, but at least one of the co-authors should commit to a detailed look at the grammar before it is resubmitted. It is often much harder for the primary author to find them.

*LdM: We have re-worked several of the parts of the discussion, gone through the text very carefully looking for grammar and spelling mistakes. We hope that the resulting changes are suitable.*

Technical comments

13 – missing comma before "and"

*LdM: done*

17 - "report" used twice

*LdM: replaced reported with found.*

32 - "allocation" used twice

*LdM: Changed to which we henceforth call ``carbon allocation''.*

52-55 – it is difficult for the reader to understand and quantify the assertion that something "appears" to be the case by looking at a figure in an IPCC Assessment – Did the Chapter assess the degree that TCRE was pathway dependent? I do not believe that was a conclusion of that chapter and went back to that figure and saw, what I would say is "negligible" pathway dependence when contextualized against the overall uncertainties.

*LdM: We have re-written this and we now include the reference to Allen et al 2009, https://www.nature.com/articles/nature08019 :*

*"It is long established that the relationship between cumulative emissions and peak warming is insensitive to the emission pathway, either in the timing of emissions or the peak emission rate (Allen et al. 2009). More recently, figure 5.31 of Canadell et al. (2021) shows negligible pathway dependence between the cumulative carbon emissions and the global mean temperature change for several projections."*

67 – need "projected" before "increased"

*LdM: added*

68 – "forecasting" should be "projected"

***LdM: changed***

69 – I am not sure what makes this statement "On the other hand"

***LdM: removed on the other hand***

77 – "simulat" is used twice

***LdM: replaced first one with generate***

77-79 – While CMIP6 indeed had data standards was not a requirement that ESMs "meet certain model quality" standards other the ones provided. As such "including" should be "were".

***LdM: changed***

80 – "remove second "a drift in the"

***LdM: done***

91 – Whether SSP5-8.5 is still "feasible" as a future projection is debatable. I would remove the word.

***LdM: replaced "****highest feasible****" with "****extremely high".***

114 – Why is "This allows us to maintain model democracy" a goal? Shouldn't we want to chose the model(s) that is structurally best? Why should models that have been falsified for ECS be considered at all? I believe the argument here is that the GWL approach accounts for ECS bias to make use of other dimensions of potential usefulness such as the carbon cycle.

***LdM: We have removed the model democracy comment, and this now reads: "They allow us to generate policy relevant assessments while exploiting the full ensemble of CMIP6 models" Please see our opening comment above for more details on why it is important to exploit the full ensemble of CMIP6 models.***

115-119 – This paragraph is highly repetitive with the first sentence misdirected. Suggest removing "The 2, 3 and 4 ◦C GWLs were chosen because"

***LdM: Along with contributions from reviewer #1, this paragraph has been reworked.***

123 – I believe "climate" is needed before "sensitivity"

***LdM: added***

126-131 – I am unclear as to how each of these terms relates quantitatively to the net carbon flux across the atmos-land interface… is that NBP? If so, that should be made explicit.

*LdM: We explicitly added this to the text.*

130 – as a flux, NBP cannot be a "prognostic variable" in the common usage (e.g. all the carbon pools) but is rather a "diagnostic variable"

*LdM: changed*

163 – "response" is necessary after "change"

*LdM: added*

217 – add comma before "and"

*LdM: added*

223,224 – "allocation" should be "allocated"

*LdM: changed*

265 – "correct" is an odd word to use here, as it connotes the potential for the analysis to be in error, whereas I believe "appropriate" is intended.

*LdM: change*

274 – is this "total carbon" the total or just the change from preindustrial? If the former, than the total in the control should be added. If the latter, than "change" should be added after "carbon"

*LdM: Changed here and elsewhere.*

299 – remove "we can assume that"

*LdM: done*

302 – again, "total carbon" is used where I believe "change" is intended as they are associated with "allocation" of emissions.

*LdM: Changed here and elsewhere.*

309 – I don't know what "is likely to be not correlated with ECS." Means… either it is correlated or it is not based on the statistic the authors define.

*LdM: Changed along with comments from reviewer #1.*

310-312 – I do not see the value of this paragraph and suggest it be removed, along with the "hollow" symbols in figure 5.

*LdM: Done.*

333 – "Due to results like these, it is widely thought that" is not a scientific statement. To what past hypotheses, assumptions, conclusions and assessments are the authors referring?

*LdM: We have re-written this as "In that figure, all five projections show a strong correlation between CO2 emissions and warming, all projections overlap at the same cumulative CO2 emissions and there are no clear differences between scenarios for the same cumulative CO2."*

367 – The statement, "This highlights the significant role that ECS plays in the uncertainty of warming projections." Suggests that the high ECS models should be considered as an uncertainty in the target rather than a deficiency in the model development process that requires fixing. Rather the point is that the willingness of some modeling centers to contribute such easily falsifiable climate responses makes for a fundamental challenge in interpretation of other aspects of the models and requires the translation into GWL.

*LdM: As we discussed in our opening comment above, ECS is only one aspect of model quality and there is no reason to expect a model's carbon cycle to be related to the model's ability to represent cloud feedbacks. By including as many models as possible, we can have a fairer representation of the state of the art in climate modelling, and benefit from a larger phase space of modelled carbon cycles.*

370-399 – This section is fairly superficial and speculative and could be removed or placed in the introduction for context.

*LdM: This section was requested by reviewer #1 and well received by them in this round. It seems appropriate to use the discussion section to highlight specific results and speculate around the topic.*

400-414 – Again, these points were made earlier in the analysis and don't seem to add anything.

*LdM: This section has been condensed and moved to the Limitations and possible extensions section. The reasoning is that a short dedicated discussion here would be useful to other groups who would want to generate their own GWL study. We have simplified this section and added some advice.*

444-445 – I don't know why the authors would say "…whereas the ocean heat content anomaly is less widely accepted outside earth system sciences." As an argument against choosing a more appropriate/objective metric than GWL if they think that is the case. It is an odd statement from a science perspective.

*LdM: Agreed! Not sure how that made it into the manuscript. I was trying to say something like "The GWLs are current defined in several international policy documents using the surface temperature, not the full volume-weighted mean of the ocean.", but I've just removed the entire sentence.*

456 – Why is the response in SSP3-7.0 "may have a noticeably different warming response to CO2"… shouldn't this already be known as part of the analysis above? What about the response was different? Perhaps a statement should be made about this in the discussion of Figure 4.`

*LdM: Rephrased it here. It's worth noting that we do highlight SSP3-7.0 in the results section as the most extreme results in the analysis, but we have added a sentence to further highlight this. However, I don't think that the discussion of SSP3-7.0 should be moved to the results section, as it is currently written as an interpretation of the results.*

490 – "on the way down as it did on the way up." Is not clear – I believe the authors are referring to reversibility and/or overshoot experiments, but both SSP1-1.9 and SSP1-2.6 include negative CO2 emissions.

*LdM: Rephrased to: "The multi-model mean of both SSP1 scenarios shows signs of recovery and carbon drawdown, but no datasets in this analysis drop below the 2 degree C GWL threshold. In future versions of this work, it would be interesting to examine the reversibility of carbon allocation with negative emission forcing scenarios. More generally, extension simulations beyond 2100 would be valuable for studying how patterns of carbon allocation change as emissions decline past net zero."*

502-505 – I do not understand what is being advocated with these sentences, "While the impact of ensemble bias is a small effect here, the multi-model means could have had a much wider range of mean ECS values between scenario groups. In the future, any investigation using the multi-model means needs to be careful with handling the equilibrium climate sensitivity bias of the ensemble. Two ensembles constituted of differing sets of models may not always be directly comparable." How can one infer that ensemble bias is "small" without having performed the data denial test of removing some of the available members to assess the effect? How could the mean "have had a much wider range of mean ECS" when that range is already outside of the assessed ECS range? In what way do people need to be "careful"? The section should be removed or rewritten with a clear recommendation in mind.

*LdM: We were lucky that our five ensembles all had a similar mean ECS. I've re-written this to be:*

> *"In this work, we attempt to maximise the number of models. ScenarioMIP's flexible*

*contributions means that each scenario's ensemble is composed of a different set of models, as shown in Table 1. This diversity results in a different mean ECS for each scenario. We were fortunate that the range of the mean ECS values was only 0.21 ∘C, despite for instance SSP1-1.9 containing significantly fewer models that the other scenarios. A different set of models could conceivably result in a wider range of mean ECS values between scenarios, which would impact the warming rates at the same CO2 concentrations, making interpretation more challenging and potentially introducing bias in the conclusions. In future investigations using CMIP6 multi-model means, the mean equilibrium climate sensitivity of each ensemble should be included alongside the analysis as two ensembles constituted of differing sets of models may not always be directly comparable."*

507 – "that" should be "how"… everyone expected that the allocations would differ. The value of the present study is in quantifying those difference and assessing the implications.

*LdM: Changed "we have shown that" to "we have quantified how".*

I appreciate the authors' work to respond to my concerns (especially the cross-validation against published studies in the results and discussion sections) and improve the quality and readability of this manuscript. I have no further general questions but one major request, to simplify the languages used in this manuscript.

The following minor revision suggestions (marked in italic format) are based on the line number in the manuscript version with tracking changes and hopefully can help authors to find directions about how to squeeze out unnecessary wordings, but authors shall double check their writings, especially the newly added content in discussion section:

*LdM: Thanks, we will endeavour to resolve these issues in this round. We appreciate the help with these suggestions. They do streamline the text. Some of the authorship team are dyslexic so we try to use short simple sentences instead of a longer ones and active voice instead of passive. This does not forgive the grammar errors and we endeavour to correct them. However, I'm happy to take these revisions and include them in a way that is suitably readable.*

Line 66: "There may also be a flux of fossil fuels directly into the ocean or land surface via for instance fossil fuel extraction and other leaks (Roser and Ritchie, 2022), but these are not generally included in Earth system models."

can be simplified.

"There may also be a direct flux of fossil fuel extraction and other leaks (Roser and Ritchie, 2022) into the ocean or land surface, but are not included in Earth system models."

*LdM: Changed*

Line 71: "Figure 5.31 of Canadell et al. (2021) shows the cumulative carbon emissions against global mean temperature change for several projections. That figure shows a strong correspondence between emissions and warming which appears to be scenario independent. "

These two sentences can be merged.

"Figure 5.31 of Canadell et al. (2021) shows the cumulative carbon emissions against global mean temperature change for several projections, but the strong correspondence between emissions and warming appears to be scenario independent."

*LdM: Changed*

Line 83: "For instance, atmospheric carbon allocation is 30% in SSP1-1.9 of the carbon remaining in the atmosphere in the year 2100, but in SSP5-8.5, that value is 62%. While the land and ocean carbon uptake are expected to remain approximately equal, the uncertainty is much larger for the

land carbon sink than the ocean. In the land, some of the uncertainty is due to the balance of increased land carbon accumulation in the high latitudes and loss of land carbon in the tropics (Canadell et al., 2021). Further uncertainty arises from the challenges of forecasting the water cycle, including droughts that reduce carbon absorption potential of the land surface. On the other hand, the ocean $CO_2$ sink is strongly dependent on the emissions scenario. This absorption of carbon into the ocean reduces the mean global buffering capacity and drives changes in the global ocean's carbonate chemistry (Jiang et al., 2019; Katavouta and Williams, 2021). These projections are based on data from the Coupled Model Inter-comparison Project (CMIP), and the most recent CMIP round, CMIP6, is described in sec. 1.1."

can be simplified.

"For instance, projected 2100 atmospheric carbon allocation from CMIP6 is 30% in SSP1-1.9 but rises to 62% in SSP5-8.5. While the land and ocean carbon uptake are expected to remain approximately equal, the uncertainty is much larger for the land carbon sink than the ocean. Uncertainty from the land sink is a tradeoff between the accumulated land carbon in the high latitudes and loss of land carbon in the tropics (Canadell et al., 2021) and the challenges of forecasting the water cycle, including droughts that reduce carbon absorption potential of the land surface. On the other hand, continuous absorption of carbon into the ocean reduces the mean global buffering capacity and drives changes in the global ocean's carbonate chemistry (Jiang et al., 2019; Katavouta and Williams, 2021), building a strong dependency on the choice of scenarios."

*LdM: Changed.*

Line 98: "Earth System models (ESMs) are one of the main tools to study the climatic impact of the combustion of fossil fuels, and they are the only tools capable of projecting the future coupled carbon-climate system. The Sixth Coupled Model Inter-comparison Project (CMIP6) (Eyring et al., 2016) is the most recent in a series of global efforts to standardise, share and study ESM simulations. To participate in CMIP6, models must simulate a suite of standard simulations and meet certain model quality and data standards. These standard simulations (also known as the Deck) include a pre-industrial control, a historical simulation, a gradual 1% $CO_2$ growth experiment and a rapid 4x$CO_2$ experiment. The quality requirements include a drift in the air-sea flux of $CO_2$ of less than 10 PgC per century, and a drift in the global volume mean ocean temperature of less than 0.1 degrees per century (Jones et al., 2011; Eyring et al., 2016; Yool et al., 2020)."

can be shortened.

"Earth System models (ESMs) are the only tools capable of projecting the future coupled carbon-climate system. The Sixth Coupled Model Inter-comparison Project (CMIP6) (Eyring et al., 2016) is the most recent global effort to standardise, share and study ESM simulations. CMIP6 designed standard simulation protocols (also known as the Deck) include a pre-industrial control, a historical simulation, a gradual 1% $CO_2$ growth experiment and a rapid 4x$CO_2$ experiment. For quality assurance, only results with a global drift per century lower than 10 PgC in the air-sea $CO_2$ flux and lower than 0.1 degrees in the volume mean ocean temperature are accepted (Jones et al., 2011; Eyring et al., 2016; Yool et al., 2020)."

*LdM: Changed*

Line 108: "In order to make projections of the future anthropogenic climate drivers, multiple scenarios were proposed in the ScenarioMIP project to cover a wide range of potential futures. ScenarioMIP expands upon the CMIP6 core simulations and multiple scenarios are available for modellers to use to generate simulations (O'Neill et al., 2016). We include the scenarios: SSP1-1.9, SSP1-2.6, SSP2-4.5, SSP3-7.0 and SSP5-8.5 (O'Neill et al., 2016; Riahi et al., 2017). Scenario names in CMIP6 are comprised of a general future pathway (SSP1-SSP5) followed by an estimate of the radiative forcing at the year 2100 in units of Wm−2. These scenarios cover a wide range of possible futures, including sustainable development in the SSP1-1.9 and SSP1-2.6 scenarios. The intermediate emissions scenario or "middle of the road" pathway in SSP2-4.5 extrapolates historic and current global development into the future with a medium radiative forcing by the end of the century. The regional ri90 valry scenario, SSP3-7.0, revives nationalism and regional conflicts, pushing global issues into the background and resulting in higher emissions. Then finally, the enhanced fossil fuel development in SSP5-8.5 is a scenario with the highest feasible fossil fuel deployment and atmospheric CO2 concentration (Riahi et al., 2017)."

can be more succinct.

"ScenarioMIP expands upon the CMIP6 core simulations with multiple scenarios of the future anthropogenic climate drivers to cover a wide range of potential futures (O'Neill et al., 2016). Scenario names in CMIP6 are a general future pathway (SSP1-SSP5) followed by an estimate of the radiative forcing at the year 2100 in units of Wm−2. These scenarios (O'Neill et al., 2016; Riahi et al., 2017) include sustainable development scenarios SSP1-1.9 and SSP1-2.6, the intermediate emissions scenario SSP2-4.5 with a medium radiative forcing by the end of the century, the regional rivalry scenario, SSP3-7.0 pushing global issues into the background and the enhanced fossil fuel development in SSP5-8.5 with the highest feasible fossil fuel deployment and atmospheric CO2 concentration (Riahi et al., 2017)."

***LdM: Changed***

Line 121: "Given the same rise in atmospheric CO2 concentration, each ESM will warm by a different amount due to the significant structural and parametric differences between models. The Equilibrium Climate Sensitivity (ECS) is a measure of this sensitivity to CO2. The ECS is given in ◦C Celsius and represents the long-term near-surface air temperature rise that is expected to result from a doubling of the atmospheric CO2 concentration once the model has reached equilibrium. In effect, the ECS is an indicator for how much warming occurs in a model with a doubling of CO2. The most recent 5-95% assessed natural ECS range was between 2 ◦C and 5 ◦C, the likely ECS range was 2.5 - 4 ◦C, and the most likely value was 3 ◦C (Arias et al., 2021, TS6). The wide spread ECS values in climate models is one of the causes of uncertainty for the timing of when forecasts reach certain warming levels. The "allowable emissions" that keep global temperature rise within policy targets are equally impacted (United Nations Treaty Collection, 2015). This has been exacerbated in the latest round of CMIP, as the CMIP6 generation of ESMs has a broader range of sensitivities than previous generations. Several CMIP6 models have a stronger response to atmospheric carbon than any CMIP5 model, and many sit above the likely ECS range (Arias et al., 2021, TS6.)"

can be more succinct.

"The Equilibrium Climate Sensitivity (ECS) is a measure defined as the near-surface air temperature rise in ∘C Celsius from a doubling of the atmospheric CO2 concentration once the model has reached equilibrium. The wide spread ECS values in climate models is one of the indicators of uncertainty for the timing of when forecasts reach certain warming levels. Due to the demand to keep global temperature rise within policy targets (United Nations Treaty Collection, 2015), more scenarios based on "allowable emissions" exacerbated a broader range of ECS in CMIP6 models than previous generations. Several CMIP6 models have a stronger response to atmospheric carbon than any CMIP5 model, and many sit above the likely ECS range. The most recent 5-95% assessed natural ECS range was between 2 ∘C and 5 ∘C, the likely ECS range was 2.5 - 4 ∘C, and the most likely value was 3 ∘C (Arias et al., 2021, TS6)."

*LdM: I've implemented this, but with some changes. In particular, this sentence doesn't make sense to me: "Due to the demand to keep global temperature rise within policy targets \citep{Paris2015}, more scenarios based on ``allowable emissions'' exacerbated a broader range of ECS in CMIP6 models than previous generations." I also re-wrote the first sentence as it introduces the  basic elements for non-experts. The resulting paragraph is:*

*"Given the same rise in atmospheric CO2 concentration, each ESM will warm to a different temperature due to the structural and parametric differences between models. The Equilibrium Climate Sensitivity (ECS) is a measure defined as the near-surface air temperature rise in ∘C from a doubling of the atmospheric CO2 concentration once the model has reached equilibrium. In nature, the 5-95% confidence range of ECS was between 2 ∘C and 5 ∘C, the likely ECS range was 2.5 - 4 ∘C, and the most likely value was 3 ∘C (Arias et al., 2021, TS6.). In ESMs, the spread of ECS values is one of the causes of uncertainty in the timing of when projections reach certain warming levels. Similarly, the uncertainty in the "allowable emissions" that would keep global temperature rise within policy targets are also impacted (United Nations Treaty Collection, 2015). This uncertainty is exacerbated in CMIP6 as it has a broader range of ECS values than previous generations and several CMIP6 models are outside the likely ECS range."*

Line 140: "Climate change policy can often focus on the climate at specific target years, like 2050 or 2100 (United Nations Treaty Collection, 2015; IPCC, 2021a). However, due to the wide range of ECS values in ESMs, this can mean that ensembles at the year 2100 are composed of a set of models with significantly different behaviours. This wide range in the temperatures and warming rates at a given point in time has knock-on effects on feedbacks and may inhibit the realism and representivity of the ensemble 110 multi-model mean (Hausfather et al., 2022; Swaminathan et al., 2022). Instead of specific target years, we can alternatively focus on model behaviour at specific Global Warming Levels (GWL), such as 2 ∘C, 3 ∘C or 4 ∘C of warming relative to the pre-industrial period. By investigating the system's behaviour at specific warming levels instead of target years, we can account for the impact of climate sensitivity and make policy relevant assessments while still exploiting the full ensemble of CMIP6 models. This allows us to maintain model democracy, even in a so-called "hot model" ensemble. The 2, 3 and 4 ∘C GWLs were chosen because the 2 ∘ C GWL is a key target set in the 2015 Paris Agreement (United Nations Treaty Collection, 2015) and thought to be a threshold for potentially dangerous climate change."

Need to be polished.

"Climate change policy focuses on the climate at specific target years, like 2050 or 2100 (United Nations Treaty Collection, 2015; IPCC, 2021a). However, due to the wide range of ECS values in ESMs,

models with significantly different behaviours projected wide range of warming rates at a given point in time, which has knock-on effects on carbon feedback and reduce realism and representativeness of the multi-model ensemble mean (Hausfather et al., 2022; Swaminathan et al., 2022). Instead of specific target years, we alternatively focus on model behaviour at specific Global Warming Levels (GWL), including 2 ◦C, 3 ◦C or 4 ◦C of warming relative to the pre-industrial period for policy relevant assessments while still exploiting the full ensemble of CMIP6 models. This allows us to maintain model democracy, even in a so-called "hot model" ensemble. The 2 ◦ C GWL is a key target set in the 2015 Paris Agreement (United Nations Treaty Collection, 2015) and thought to be a threshold for potentially dangerous climate change."

*LdM: I have re-written this section, but I find that it is much clearer and understandable as two sentences. The revised text is:*

> *"Climate change policy has a tendency to focus on the climate at specific target years, such as 2050 or 2100 (United Nations Treaty Collection, 2015; IPCC, 2021a). However, due to the diversity of ECS values in CMIP6, the ensemble will project a wide range of warming rates and surface temperatures at a given point in time. This wide range of behaviours has knock-on effects on climate feedbacks and may inhibit the realism and representativeness of the ensemble's multi-model mean (Hausfather et al., 2022; Swaminathan et al., 2022). On the other hand, this more comprehensive range of responses is valuable in exploring carbon-climate processes that are of direct relevance to policy. Instead of specific target years, we focus on three specific Global Warming Levels (GWL). These are 2 ◦C, 3 ◦C or 4 ◦C of warming relative to the pre-industrial period. They allow us to generate policy relevant assessments while exploiting the full ensemble of CMIP6 models. Not only does the GWL methodology mirror the policy discourse surrounding the policy targets, it is also largely independent of the choice of future emissions scenario as the world largely looks the same at 2 ◦C, no matter how we get there (Hausfather et al., 2022). In addition, GWL bypasses the need to select or weight CMIP6 models as each model provides distinct and relevant information, so the full CMIP6 ensemble can be used (Hausfather et al., 2022). The three GWLs were chosen because the 2 ◦C GWL is a key target set in the 2015 Paris Agreement and thought to be a threshold for potentially dangerous climate change (United Nations Treaty Collection, 2015). The 3 ◦C GWL is the warming level that current nationally determined emission policies will realise for the year 2100 assuming a median climate sensitivity (United Nations Environment Programme, 2019). Finally, the 4 ◦C GWL is a low likelihood but high impact outcome if climate sensitivity is higher than median values or emission reductions and climate policy break down (World Bank, 2012)"*

Line 186: "The NBP is an prognostic variable calculated" shall be "a prognostic variable"

*LdM: Changed .*

Line 297: "The left side shows the percentage allocation, and the right side shows the totals in PgC." Move figure explanation to the figure caption.

*LdM: Done*

Line 299/300: "More carbon is allocation" shall be "More carbon is allocated"

**LdM: Done**

Line 300: revise to "is allocate"

**LdM: done**

Line 335: "Figure 2 only shows the multi-model means, not single models. This means that multi-model means that do not reach the GWL are not included in this figure."

Move this to the corresponding figure caption.

**LdM: I disagree, this is clearer if it stays in the text.**

Line 387" "This figure includes a pair of panes for each experiment scenario. For each pair, the top pane is the cumulative carbon in PgC and the bottom pane shows the percentage."

Move to the figure caption.

**LdM: done**

Line 413: "This results in the saw-tooth pattern on the right of this figure. However, this saw-tooth pattern does not appear on the left side of the figure, as the ratios of carbon allocation between land, ocean and atmosphere at a given GWL are not dependent on ECS. "

can be simplified.

"However, the ratios of carbon allocation between land, ocean and atmosphere at a given GWL are not dependent on ECS, since the pattern is relatively smooth comparing to the C emission."

**LdM: changed.**

Line 426: "Firstly, at a given GWL, higher emission scenarios have a higher atmospheric fraction. In effect, the SSP5-8.5 scenarios have a higher atmospheric fraction than SSP1-1.9 and SSP1-2.6 scenarios, even at the same GWL. Similarly, higher emission scenarios have a smaller land fraction, while the ocean fraction is similar across scenarios at the same GWL. Secondly, warmer GWLs have a larger atmospheric fraction than cooler GWLs. Thirdly warmer GWLs have a smaller land fraction than cooler GWLs. Finally, the ocean fraction is relatively consistent between GWLs and scenarios."

The expression shall be simplified.

"At a given GWL, higher emission scenarios have a higher atmospheric fraction, but a lower land fraction and a relatively consistent ocean fraction. When comparing the allocation fractions from different GWLs, warmer GWLs have larger atmospheric fractions, lower land fractions and consistent ocean fractions than colder GWLs."

**LdM: added - This is a great improvement, thanks.**

Line 432: "The data from fig.4 is re-framed in fig.5 as a series of scatter plots. In this figure, each row represents a different scenario, and each column is a different dataset. These datasets are: the GWL threshold year, the total carbon allocated, the carbon allocation for each domain and the fractional carbon allocation to each domain. The y-axis shows the model's ECS, and each point is a different GWL, where the squares are 2 ◦C GWL, the circles are 3 ◦C GWL, and the triangles are 4 ◦C GWL." Move to the figure caption.

***LdM: Done.***

Line 437: "For each group of data, the line of best fit is shown and the absolute value of the fitting error (Err) of the slope (M) over the slope is shown in the legend, as Err/M. The fitting error, Err, here is the standard error of the estimated gradient under the assumption of residual normality. This value indicates whether the slope crosses the origin within the 95% confidence limit. If the uncertainty on the slope is greater than the slope itself (and Err/M exceeds unity), then we can assume that the fit is not statistically significant. All groups with three models or fewer that reach the GWL were excluded as this is not enough data points to draw meaningful conclusions.

The goal of this figure is to highlight in broad strokes the ways that ECS interacts with carbon allocation in these models. In most of the fits, the data and the ECS are inversely correlated such that lower ECS models have higher values. This appears to be true for GWL year, total carbon and the individual total carbon fields in the atmosphere, ocean and land. The GWL threshold year and the total carbon allocations both have all absolute Err/M values lower than unity and as such are both related to ECS. In both the ocean and the atmosphere's total carbon, the absolute value of Err/M is always smaller than one. This means that the total carbon in both the ocean and the atmosphere are linked to ECS with 95% confidence. However, this is not the case for the ocean or the atmosphere's carbon allocation as a percentage and in many cases absolute Err/M is greater than unity. This means that we can not say that the fraction of carbon allocated to the ocean or to the atmosphere is related to the ECS with 95% confidence. Similarly, this absolute Err/M ratio is not consistently below unity for the land ensembles at all GWLs. This implies that the total or percentage land carbon allocation is likely to be not correlated with ECS."

This part is way too verbose. I suggest to revise:

"For each group of data, the line of best fit is shown and the absolute value of the fitting error (Err, the standard error of the estimated gradient under the assumption of residual normality) over the slope (M) is shown in the legend, as Err/M. This value indicates whether the slope crosses the origin within the 95% confidence limit (< 1) or not (> 1). All groups with three models or fewer that reach the GWL were excluded as not enough data points to draw meaningful conclusions.

GWL year, total carbon and the individual total carbon allocation fractions are inversely correlated to ECSs. The GWL threshold year and the total carbon allocations both have all absolute Err/M values lower than unity and as such are both related to ECS. The total carbon in both the ocean and the atmosphere are linked to ECS, as their Err/M are smaller than 1. However, the correlations between carbon allocation fraction of the ocean or the atmosphere and ECS are not statistically significant. For land, both total carbon sink and allocation fraction are not significantly correlated to ECS at all GWLs."

*LdM: Thanks – that is simpler and clearer.*

---

## Author Response (AR3)

**Scenario Choice Impacts Carbon Allocation Projection at Global Warming Levels - Authors response to Anonymous referee #1 – 2023-10-04**

Hi,

Thanks to the referee and the editor for their on-going work with this manuscript. We appreciate you taking the time to review the paper again. The changes that you've suggested made have improved the manuscript. We went through the three specific comments and addressed them. Then we did a thorough readthrough for grammar, spelling, typesetting and consistency. While few changes were requested, many changes were made as can be seen in the track-changes file.

Kind regards,

Lee de Mora and the Authorship team.

---

## Author Response (AR4)

**Scenario Choice Impacts Carbon Allocation Projection at Global Warming Levels – Final Authors response**

Hi,

Thanks to the referee and the editor for all the hard work this manuscript. We have finalised our files in the submitted document. The only change is the addition of the link to the video abstract. I also include it here for your convinience:

Scenario Choice Impacts Carbon Allocation Projection at Global Warming Levels - Video Abstract

Kind regards,

Lee de Mora and the Authorship team.